# Symmetry-Constrained Causal Partial Identification

## Abstract

We present a novel framework for using knowledge of data symmetries to sharpen bounds in causal *partial identification (PI)*. The causal effect of the treatment $X$ on outcome $Y$ is generally not identifiable from observational data alone if their common causes, also known as confounders, are unobserved. PI entails estimating bounds on such treatment effects by solving a constrained optimization problem that encodes different assumptions imposed on data generation. PI has use in many application domains where such bounds are sufficient to inform policy decisions, even if the treatment effect itself is not identifiable. We show that knowledge of symmetries in data generation—formalized as invariance under transformation groups—provides additional constraints that tighten these bounds. We operationalize this insight through two approaches: (i) adding explicit invariance error constraints to existing PI methods, and (ii) applying symmetry-preserving *data augmentation (DA)* as a pre-processing step. Under a linear Gaussian model, we show that the later yields bounds that provably valid (containing the true causal effect), sharper (smaller identified sets), and more robust (lower worst-case error). The key mechanism being that randomized symmetry transformations introduce exogenous variation in $X$ that cannot be attributed to confounding, thereby reducing ambiguity in the identified set. Experiments on synthetic and real data validate our approach. More broadly, our findings establish known data symmetries—ubiquitously employed in DA for variance reduction—can be repurposed as a principled tool for causal inference when point-identification is impossible.

## 1 Introduction

The problem of regression in machine learning aims to fit a model to observational $(X, Y)$ data that predicts outcome $Y$ from treatment $X$. Improving the generalization of such predictors to unlabeled samples of $X$ often requires regularization techniques like *data augmentation (DA)* (Vapnik, 1998; Shorten & Khoshgoftaar, 2019; Lyle et al., 2020). However, such predictive models are generally not causal: the statistical relationship between $X$ and $Y$ may be driven by unobserved common causes, i.e. confounders, rather than the true influence of $X$ on $Y$. The gold standard for eliminating confounding is direct intervention, i.e. explicit randomization of $X$ during data generation (Peters et al., 2017; Pearl, 2009). Since these are often inaccessible, a common workaround is to correct for confounding via auxiliary variable (Zhang et al., 2023). However, these too may be insufficient to recover the causal effect (Kilbertus et al., 2020), or scarce in many applications (Akbar et al., 2025).

In such cases, identifying the true causal effect is not possible from observational data alone. *Partial identification (PI)* offers a principled alternative by computing bounds guaranteed to contain the true effect (Padh et al., 2023)—often sufficient for decision-making even without point-identification.. These bounds are obtained by solving an optimization problem whose constraints encode assumptions about how the data were generated. The informativeness of the bounds depends entirely on the strength and structure of these constraints.

This paper introduces known data symmetries—formalized as invariance under transformations of $X$—as a new source of constraints for PI. Such symmetries are ubiquitous in scientific and causal modeling, ranging from geometric stabilities in physical systems (Bronstein et al., 2021; Satorras et al., 2022) to semantic invariance in natural language (Veitch et al., 2021a) and permutation in-

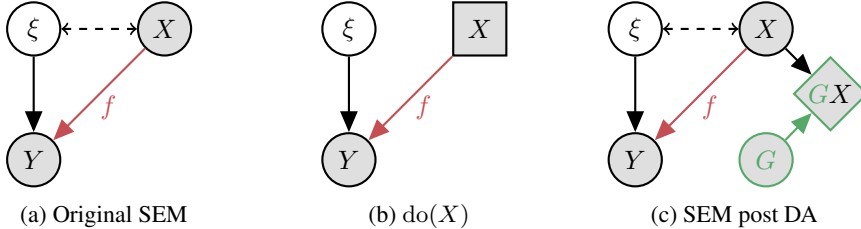

(a) Original SEM        (b) $\mathrm{do}(X)$        (c) SEM post DA

Figure 1: Graphs of respective SEMs. ($a$) The original SEM from Eq. (1) with confounded $(X, Y)$. ($b$) The original SEM post intervention on $X$. ($c$) The original SEM post DA application.

variance in exchangeable data (Veitch et al., 2021b). We demonstrate that enforcing this structural knowledge restricts the set of plausible causal hypotheses, effectively pruning the identified set. We operationalize this insight in two complementary ways:

1. **Explicit constraints via invariance error (Section 4.1)**: we propose bounding the invariance error under specified transformations as an explicit constraint for improving PI.

2. **Implicit constraints via DA (Section 4.2)**: we show the effectiveness of a simple symmetry-preserving DA based pre-processing step before running off-the-shelf PI solvers.

The first approach gives strict improvement in PI by construction, is amenable to modern Monte Carlo, gradient-based solvers, is not restricted to any specific hypothesis class and allows tunability to easily handle symmetry miss-specification. The later provides a cheap tool that can be composed with black-box PI solvers, offering guarantees under the linear Gaussian regime with well-specified symmetries. Our methods are simple to apply, compatible with existing solvers, and re-purpose the rather pervasive ML tool of data symmetries as a practical approach for strengthening causal conclusions when point-identification not possible.

## 2 PRELIMINARIES

### 2.1 STATISTICAL VS. CAUSAL INFERENCE

Consider random treatment $X$, outcome $Y$ taking values in $\mathcal{X} \subseteq \mathbb{R}^m$, $\mathcal{Y} \subseteq \mathbb{R}$ respectively. The function $f \in \mathcal{H} := \{h : \mathcal{X} \to \mathcal{Y}\}$ defines their causal relationship via a *structural equation model (SEM)*

$$Y = f(X) + \xi, \qquad \mathbb{E}[\xi] = 0. \qquad (1)$$

We want to estimate $f$ given a dataset $\mathcal{D} := \{(\boldsymbol{x}_i, y_i)\}_{i=0}^{n}$ of $n$ samples from the distribution $\mathbb{P}_{X,Y}$.

With the assumption $X \perp\!\!\!\perp \xi$, we have $\mathbb{E}[Y \mid X = \boldsymbol{x}] = f(\boldsymbol{x})$ in Eq. (1). *Statistical inference* entails identifying precisely the Bayes optimal predictor $\mathbb{E}[Y \mid X = \boldsymbol{x}]$ from $\mathcal{D}$ by minimizing an empirical version of the *statistical risk* over hypotheses $h \in \mathcal{H}$ for some proper, convex loss $\ell : \mathbb{R} \times \mathbb{R} \to \mathbb{R}_+$,

$$R_{\mathrm{erm}}(h) := \mathbb{E}[\ell(Y, h(X))]. \qquad (2)$$

Then, for a sufficiently rich hypothesis class, the minimizer $h_{\mathrm{erm}}$ gives an unbiased estimation of $f$.

However, the residual $\xi$ in Eq. (1) may generally be correlated with $X$, i.e., $\mathbb{E}[\xi \mid X] \neq 0$, so that the conditional $\mathbb{E}[Y \mid X = \boldsymbol{x}]$ now gives a biased estimate of $f(\boldsymbol{x})$ (Pearl, 2009; Peters et al., 2017). This correlation arises due to unobserved common causes of $X$ and $Y$, known as *confounders*. We say that $X$ and $Y$ are confounded and refer to the resulting bias as the *confounding bias* (Pearl, 2009). *Causal inference* entails adjusting for this bias to identify $f$, or at the very least account for it by finding bounds on $f$ should identification not be possible. Both approaches are outlined below.

### 2.2 INTERVENTION FOR CAUSAL EFFECT IDENTIFICATION

We can make $X$ and the residual $\xi$ uncorrelated via an *intervention* $\mathrm{do}(X := X')$ that explicitly sets $X$ to some independently sampled $X'$ in Eq. (1) during data generation. The induced distribution, referred to as the *interventional distribution*, is represented by $\mathbb{P}_{X,Y}^{\mathrm{do}(X := X')}$. We use the shorthand

notation $\mathrm{do}(X)$ for an intervention where $X' \sim \mathbb{P}_X$, under which the objective from Eq. (2) now defines the *causal risk* (Kania & Wit, 2023; Vankadara et al., 2022; Janzing & Schölkopf, 2018b) as

$$R_{\mathrm{erm}}^{\mathrm{do}(X)}(h) := \mathbb{E}^{\mathrm{do}(X)}[\ell(Y, h(X))]. \tag{3}$$

The target estimand of Eq. (3) is the *average treatment effect (ATE)* $\mathbb{E}^{\mathrm{do}(X:=\boldsymbol{x})}[Y \mid X = \boldsymbol{x}]$ which equals $f(\boldsymbol{x})$ for the SEM under consideration in Eq. (1). Minimizers of Eq. (3) therefore give an unbiased estimation of $f$. To better capture the *estimation error* for a candidate hypothesis $h \in \mathcal{H}$, we use the *causal excess risk* (Vankadara et al., 2022) by removing irreducible noise from Eq. (3) as

$$E^{\mathrm{do}(X)}(h) := R_{\mathrm{erm}}^{\mathrm{do}(X)}(h) - R_{\mathrm{erm}}^{\mathrm{do}(X)}(f).$$

Since interventions are often inaccessible for computing the risk Eq. (3), estimating $f$ usually relies on access to the full joint distribution $\mathbb{P}$ of $(X, Y, \xi)$ via *back-door adjustment* (Xu & Gretton, 2022)

$$h_{\mathrm{adj}}^{\mathbb{P}}(\boldsymbol{x}) := \mathbb{E}_\xi[\mathbb{E}[Y \mid X = \boldsymbol{x}, \xi]], \qquad (X, Y, \xi) \sim \mathbb{P}.$$

### 2.3 Partial identification and sensitivity analysis

For unobserved noise $\xi$, identification of $f$ is generally not possible from $\mathbb{P}_{X,Y}$ alone. Nevertheless, given assumptions on the data generating process in Eq. (1), we can do *partial identification (PI)* (Padh et al., 2023) of $f$ by considering all the joint distributions $\mathbb{Q}$ consistent with said assumptions,

$$\mathcal{Q}_{\mathrm{pi}}(\mathbb{P}_{X,Y}) := \left\{ \mathbb{Q} \in \mathcal{C}_{\mathrm{pi}} \,\middle|\, \mathbb{Q}_{X,Y} = \mathbb{P}_{X,Y} \right\},$$

where the constraint set $\mathcal{C}_{\mathrm{pi}}$ encodes our assumptions. If correctly specified, $\mathbb{P} \in \mathcal{C}_{\mathrm{pi}}$ and the following set $\mathcal{H}_{\mathrm{pi}}$ of candidate hypotheses contains the true solution $f$, or the interval $\mathcal{H}_{\mathrm{pi}}(\boldsymbol{x})$ holds $f(\boldsymbol{x})$,

$$\mathcal{H}_{\mathrm{pi}} := \left\{ h_{\mathrm{adj}}^{\mathbb{Q}} \,\middle|\, \mathbb{Q} \in \mathcal{Q}_{\mathrm{pi}} \right\}, \qquad \mathcal{H}_{\mathrm{pi}}(\boldsymbol{x}) := \left\{ h_{\mathrm{adj}}^{\mathbb{Q}}(\boldsymbol{x}) \,\middle|\, \mathbb{Q} \in \mathcal{Q}_{\mathrm{pi}} \right\},$$

where $\mathcal{Q}_{\mathrm{pi}}$ is shorthand for $\mathcal{Q}_{\mathrm{pi}}(\mathbb{P}_{X,Y})$. Computing the interval $\mathcal{H}_{\mathrm{pi}}(\boldsymbol{x})$ at $\boldsymbol{x}$ is often more practical than characterizing the set $\mathcal{H}_{\mathrm{pi}}$, since it amounts to solving two constrained optimization problems as

$$\mathcal{H}_{\mathrm{pi}}(\boldsymbol{x}) = \left[ \min_{\mathbb{Q} \in \mathcal{Q}_{\mathrm{pi}}} h_{\mathrm{adj}}^{\mathbb{Q}}(\boldsymbol{x}), \ \max_{\mathbb{Q} \in \mathcal{Q}_{\mathrm{pi}}} h_{\mathrm{adj}}^{\mathbb{Q}}(\boldsymbol{x}) \right]. \tag{4}$$

In either case, we want the identified sets to (i) contain the true solution, (ii) be as small as possible.

The constraint set may also be parameterized as $\mathcal{C}_{\mathrm{pi}}(\boldsymbol{\Gamma})$ to conduct *sensitivity analyses* (Frauen et al., 2024) by varying parameters $\boldsymbol{\Gamma}$ to see how $\mathcal{H}_{\mathrm{pi}}, \mathcal{H}_{\mathrm{pi}}(\boldsymbol{x})$ evolve as assumptions are relaxed/tightened.

Lastly, since we are now discussing hypothesis sets, we define the two appropriate evaluation metrics

$$E_{\mathrm{approx}}^{\mathrm{do}(X)}(\mathcal{Q}_{\mathrm{pi}}) := \min_{\mathbb{Q} \in \mathcal{Q}_{\mathrm{pi}}} E^{\mathrm{do}(X)}\left(h_{\mathrm{adj}}^{\mathbb{Q}}\right), \qquad E_{\mathrm{worst}}^{\mathrm{do}(X)}(\mathcal{Q}_{\mathrm{pi}}) := \max_{\mathbb{Q} \in \mathcal{Q}_{\mathrm{pi}}} E^{\mathrm{do}(X)}\left(h_{\mathrm{adj}}^{\mathbb{Q}}\right).$$

The *approximation error* $E_{\mathrm{approx}}^{\mathrm{do}(X)}(\mathcal{Q}_{\mathrm{pi}})$ measures how far the target $f$ is from $\mathcal{H}_{\mathrm{pi}}$ (Brown & Ali, 2024), and the *worst-case excess risk* $E_{\mathrm{worst}}^{\mathrm{do}(X)}(\mathcal{Q}_{\mathrm{pi}})$ upper bounds the performance of the identified set $\mathcal{H}_{\mathrm{pi}}$ relative to the target $f$. Similarly, $\mathcal{H}_{\mathrm{pi}}(\boldsymbol{x})$ is evaluated using $E_{\mathrm{approx}}^{\mathrm{do}(\boldsymbol{x})}(\mathcal{Q}_{\mathrm{pi}})$ and $E_{\mathrm{worst}}^{\mathrm{do}(\boldsymbol{x})}(\mathcal{Q}_{\mathrm{pi}})$.

**Choice of constraints.** The nature and construction of $\mathcal{C}_{\mathrm{pi}}$ often depends on domain knowledge. Popular approaches involve bounding the spurious correlation between $X, Y$, including the sensitivity model by Rosenbaum (2002) which parameterizes the strength of unmeasured confounding through odds ratios, its generalization of the Marginal Sensitivity Model (MSM) by Tan (2006) that does the same using propensity scores and the partial R-squared approach by Cinelli & Hazlett (2020) bounds the proportion of variance explained by unobserved confounders. More recently, Fan et al. (2024); Guo et al. (2022) formulated the PI problem as *robust optimization (RO)* over $\mathcal{Q}_{\mathrm{pi}}$ constructed as a total variation ball around the observational distribution $\mathbb{P}_{X,Y}$, and Meresht et al. (2022) similarly uses Wasserstein constraints. An equivalent approach to modeling the confounding is to instead model the random function $f_\xi(\cdot) := f(\cdot) + \xi$ itself, also known as the *response function* (Padh et al., 2023). Hu et al. (2021) modeled these using generative adversarial networks (GANs) to then match $\mathbb{P}_{X,Y}$ in distribution. Most of these methods can also leverage auxiliary variables in addition to $X, Y$ for imposing constraints in the form of conditional independences to sharpen bounds. Of note is the instrumental variable (IV) based PI by Balke & Pearl (1997) for when $\xi$ arbitrarily influences $Y$ instead of the additive model in Eq. (1). Modern neural-network based variants for continuous, high-dimensional treatments and/or IVs are explored by Schweisthal et al. (2025); Kilbertus et al. (2020); Hu et al. (2021); Padh et al. (2023); Meresht et al. (2022); Gunsilius (2020).

## 2.4 DATA SYMMETRIES AND INVARIANCE

For finite samples, the technique of *data augmentation (DA)* is used to reduce estimation variance (Lyle et al., 2020; Chen et al., 2020) in statistical inference. This is achieved by applying random transformations $G \sim \mathbb{P}_G$ to the data, generating multiple transformed samples $(G\boldsymbol{x}_i, y_i)$ from each original sample $(\boldsymbol{x}_i, y_i) \in \mathcal{D}$, thereby increasing variability in the data for statistical risk evaluation,

$$R_{\mathrm{da+erm}}(h) := \mathbb{E}[\ell(Y, h(GX))]. \tag{5}$$

In this work we restrict ourselves to DA with respect to which $f$ is invariant (Lyle et al., 2020; Chen et al., 2020). The action of a group $\mathcal{G}$ is a mapping $\alpha : \mathcal{X} \times \mathcal{G} \to \mathcal{X}$ compatible with the group operation. Writing $\boldsymbol{g}\boldsymbol{x} := \alpha(\boldsymbol{x}, \boldsymbol{g})$ as shorthand, we say that $f$ is *invariant* under $\mathcal{G}$ (or $\mathcal{G}$-*invariant*) if

$$f(\boldsymbol{g}\boldsymbol{x}) = f(\boldsymbol{x}), \qquad \forall (\boldsymbol{g}, \boldsymbol{x}) \in \mathcal{G} \times \mathcal{X}.$$

Group $\mathcal{G}$ has a (unique) normalized Haar measure, $\mathbb{P}_G$ the corresponding distribution defined over it.

Of course one needs to have prior knowledge about the symmetries of $f$ to construct such a DA. We argue that the popularity of this modeling assumption in the DA and invariance literature (Lyle et al., 2020; Chen et al., 2020) is precisely because such symmetries are already established in many application domains. For example, when classifying images of cats and dogs we already know that whatever the true labeling function may be, it would certainly be invariant to rotations on the images. $G$ would then represent the random rotation angle, whereas $G\boldsymbol{x}$ would be the rotated image $\boldsymbol{x}$.

While DA is canonically used to mitigate finite-sample estimation variance, our focus is primarily on the infinite-sample setting, and we present Eq. (5) and subsequent theoretical results in that context. Nonetheless, increasing sample size via DA also bears on our work, a point we shall briefly discuss. Section 4.2 also makes the following assumption which characterizes many standard DA operations.

**Assumption 1** (unbiased group action). *The group action $G \sim \mathbb{P}_G$ is identity-centered, meaning*

$$\mathbb{E}[G\boldsymbol{x}] = \boldsymbol{x}, \qquad \forall \boldsymbol{x} \in \mathcal{X}.$$

**Lemma 1** (added exogenous variation with DA). *Under Assumption 1, $G$ inflates the data variance,*

$$\boldsymbol{\Delta} := \boldsymbol{\Sigma}_{GX} - \boldsymbol{\Sigma}_X \succcurlyeq \boldsymbol{0}, \qquad \text{equality iff} \quad GX = X \quad a.s.$$

*Proof.* See Appendix A.5 for the proof. $\qquad\square$

## 3 CAUSAL IMPLICATIONS OF DATA SYMMETRIES

Crucially, the random group action $G$ from Lemma 1 introduces additional *exogenous* variation in $X$ that is independent of other system variables. Consequently, Akbar et al. (2025) showed that for a $\mathcal{G}$-invariant target function $f$, the transformation $GX$ simulates a *soft* intervention on $X$—perturbing $X$ to weaken the confounding association $X \leftrightarrow \xi$ while preserving the causal mechanism $X \to Y$. To formalize how this improves point estimation of $f$, Akbar et al. (2025) employs the following linear version of Eq. (1), which also serves as the basis for our subsequent PI analysis in Section 4.2.

**Assumption 2** (a linear, Gaussian SEM). *The SEM Eq. (1) is centered, joint Gaussian with $\boldsymbol{f} \in \mathbb{R}^m$,*

$$Y = \boldsymbol{f}^\top X + \xi.$$

In this setting, the causal estimation error (excess risk) under a squared loss takes the following form:

$$E^{\mathrm{do}(X)}(\boldsymbol{h}) = \|\boldsymbol{h} - \boldsymbol{f}\|_{\boldsymbol{\Sigma}_X}^2, \qquad E^{\mathrm{do}(\boldsymbol{x})}(\boldsymbol{h}) = \left(\boldsymbol{h}^\top \boldsymbol{x} - \boldsymbol{f}^\top \boldsymbol{x}\right)^2, \tag{6}$$

where we overload the notation $\mathrm{do}(\boldsymbol{x})$ (as opposed to $\mathrm{do}(X)$) to meant the *hard* intervention $\mathrm{do}(X := \boldsymbol{x})$, i.e. fixing $X$ to a constant $\boldsymbol{x}$ during data generation. Similar formulations have been used to measure causal error (Vankadara et al., 2022; Kania & Wit, 2023; Akbar et al., 2025) or quantify confounding strength (Janzing, 2019; Janzing & Schölkopf, 2018a;b). (Akbar et al., 2025) established the following result, which directly bears on our work:

**Proposition 1** (estimation with DA (Akbar et al. (2025) lifted))**.** *For $\mathcal{G}$-inv. $\boldsymbol{f}$, Assumptions 1 and 2,*

$$0 \;\leq\; \underbrace{\frac{\kappa}{1+\kappa} \cdot \; \|\boldsymbol{\Pi}_{\boldsymbol{\Delta}}(\boldsymbol{h}_{\mathrm{erm}} - \boldsymbol{f})\|^2_{\boldsymbol{\Sigma}_X}}_{\textit{estimation error within } \mathrm{range}(\boldsymbol{\Delta})} \;\leq\; E^{\mathrm{do}(X)}(\boldsymbol{h}_{\mathrm{erm}}) - E^{\mathrm{do}(X)}(\boldsymbol{h}_{\mathrm{da+erm}}), \qquad \overset{\textit{DA orthogonal}}{\underset{}{\textit{to confounding}}}$$

$$\leq\; \|\boldsymbol{\Pi}_{\boldsymbol{\Delta}}(\boldsymbol{h}_{\mathrm{erm}} - \boldsymbol{f})\|^2_{\boldsymbol{\Sigma}_X}, \qquad \textit{eq. iff} \qquad \overbrace{\boldsymbol{\Delta} \perp \boldsymbol{\Sigma}_{X,\xi}},$$

*where $\kappa := \lambda^+_{\min}\left(\boldsymbol{\Sigma}_X^{-1}\boldsymbol{\Delta}\right) < \infty$ represents the lowest positive eigenvalue of the product $\boldsymbol{\Sigma}_X^{-1}\boldsymbol{\Delta}$.*

*Proof.* See Appendix A.3, cf. (Akbar et al., 2025, Thm. 3) for the proof. □

Essentially, for $\mathcal{G}$-invariant $\boldsymbol{f}$, DA *dominates* ERM on causal estimation—performing strictly better iff it aligns with the confounding within $X$, but never worse. Note that in Proposition 1, for the "large DA" regime, which we define as $\kappa \to \infty$, the lower-bound approaches the upper-bound, which is simply the sq.-norm of the *projection* $\boldsymbol{\Pi}_{\boldsymbol{\Delta}}(\cdot)$ of estimation bias $(\boldsymbol{h}_{\mathrm{erm}} - \boldsymbol{f})$ onto $\mathrm{range}(\boldsymbol{\Delta})$. This confirms that identification is generally not possible in this setting. Therefore the principled approach is to undertake PI of $\boldsymbol{f}$ instead of the point-estimation approach by Akbar et al. (2025).

This motivates our current work, where we leverage knowledge of symmetries in data generation to improve partial identification and/ or sensitivity analysis of $f$, as discussed in the following section.

## 4 Symmetry-Constrained Partial Identification

Our objective is to leverage symmetry knowledge to restrict the identified sets $\mathcal{H}_{\mathrm{pi}}$ and $\mathcal{H}_{\mathrm{pi}}(\boldsymbol{x})$. We give two strategies to operationalize this: (i) integrating an explicit invariance error constraint into the optimization, and (ii) inducing implicit regularization through data augmentation pre-processing.

### 4.1 Explicit constraint with invariance error

We start off by considering the most obvious approach to incorporate symmetry knowledge into PI— add an explicit invariance error constraint to any baseline PI method defined by a constraint set $\mathcal{C}_{\mathrm{pi}}$,

$$E_{\mathrm{inv}}(h) := \mathbb{E}\left[\, \|h(X) - h(GX)\|^2 \,\right],$$

$$\mathcal{Q}_{\mathrm{inv+pi}}(\mathbb{P}_{X,Y}) := \left\{\, \mathbb{Q} \in \mathcal{C}_{\mathrm{pi}} \,\middle|\, \mathbb{Q}_{X,Y} = \mathbb{P}_{X,Y}, \;\; E_{\mathrm{inv}}\left(h^{\mathbb{Q}}_{\mathrm{adj}}\right) \leq \epsilon \,\right\}.$$

**Remark 1** (sharper, robust bounds with invariance error)**.** By construction, subset inclusion holds:

$$\mathcal{H}_{\mathrm{inv+pi}} \subseteq \mathcal{H}_{\mathrm{pi}}, \qquad\qquad \mathcal{H}_{\mathrm{inv+pi}}(\boldsymbol{x}) \subseteq \mathcal{H}_{\mathrm{pi}}(\boldsymbol{x}).$$

Consequently, this guarantees that the volume of the identified set and the corresponding worst-case excess risk does not increase. Note that due to this same set inclusion, the approximation error generally cannot decrease, and may even potentially increase if $\mathcal{C}_{\mathrm{pi}}$ does not contain the true distribution $\mathbb{P}$. For $\epsilon = 0$, these metrics are equal to "large DA" regime results in Sections 3 and 4.2.

Nevertheless, whenever the baseline PI constraints $\mathcal{C}_{\mathrm{pi}}$ are valid and $E_{\mathrm{inv}}(f) \leq \epsilon$ holds, $\mathcal{Q}_{\mathrm{inv+pi}}$ guarantees validity. Furthermore, the parameter $\epsilon$ enables sensitivity analysis, allowing us to inspect how the bounds evolve as we vary the assumed invariance error in our choice of transformations $G$. Of course we can similarly use other formulations for $E_{\mathrm{inv}}$, such as ones stated in Yang et al. (2019), or restrict ourselves to a hypothesis class that follows our symmetry by design (Cohen & Welling, 2016). However, the later may be restrictive in terms of compatibility with standard PI methods.

While our experiments discuss settings where Eq. (4) for $\mathcal{H}_{\mathrm{inv+pi}}(\boldsymbol{x})$ can be solved via convex programming, we emphasize that this explicit constraint formulation is fully compatible with modern deep learning-based PI. Since the functional $E_{\mathrm{inv}}$ is differentiable and amenable to Monte Carlo evaluation, it can be readily incorporated as a regularizer in augmented Lagrangian and/or gradient-based solvers (Padh et al., 2023; Kilbertus et al., 2020; Meresht et al., 2022; Hu et al., 2021).

Despite this compatibility, incorporating the invariance error constraint still requires modifying the solver logic or objective function. This imposes an implementation burden and precludes the use of specialized or "black-box" PI software where the internal optimization is fixed. This limitation motivates our second approach—a simple data pre-processing strategy that implicitly impose symmetry

constraints by simply feeding augmented data into standard off-the-shelf PI methods. Furthermore, when modeling complex, high-dimensional data, enforcing non-convex invariance constraints during optimization is often more expensive and notoriously unstable (Schweisthal et al., 2025; Padh et al., 2023) as opposed to a simple data pre-processing step.

## 4.2 IMPLICIT CONSTRAINT WITH DA PRE-PROCESSING

We draw inspiration from IV methods, where "strong" instruments—those inducing significant exogenous variation in $X$—are known to yield sharper identification bounds compared to weak instruments (Kilbertus et al., 2020; Padh et al., 2023). This motivates our central inquiry in this section:

*Can the synthetic exogenous variation introduced by DA similarly sharpen PI?*

As with Akbar et al. (2025), which we extend now to the PI setting, the fundamental mechanism for PI sharpening is Lemma 1. Our main insight into why DA aids PI is summarized as follows:

(i) **Statistical Efficiency:** Most straightforwardly, DA grows effective data size, quelling sampling variation and finite-sample errors; key sources of uncertainty in PI (Imbens & Manski, 2004).

(ii) **Sharper Bounds:** DA adds variation in $X$ that is explicitly exogenous, and therefore cannot be attributed to confounding. This reduces ambiguity in PI, which leads to sharper bounds.

(iii) **Robust Bounds:** By perturbing spurious features, DA reduces confounding bias, centering and contracting the PI bounds around the true solution. This directly minimizes the worst-case error.

(iv) **Valid Bounds:** Crucially, $\mathcal{G}$-invariance of $f$ guarantees valid bounds with DA if $\mathcal{C}_{\mathrm{pi}}$ is valid.

We elaborate these via analysis of the linear model from Assumption 2. But first we explicitly define the composition $\mathcal{Q}_{\mathrm{da+pi}}$ of DA and PI, as well as the specific PI model that we use for our analysis,

$$\mathcal{Q}_{\mathrm{da+pi}}(\mathbb{P}_{X,Y}) \coloneqq \mathcal{Q}_{\mathrm{pi}}(\mathbb{P}_{GX,Y}).$$

**Assumption 3** (a bounded confounding sensitivity model)**.** Consider the following constraint set.

$$\mathcal{C}_{\mathrm{pi}}(\mathbf{\Gamma}) \coloneqq \left\{ \mathbb{Q} = \mathcal{N}(\mathbf{0}, \cdot) \;\middle|\; \frac{\mathrm{Var}(\mathbb{E}[\xi \,|\, X])}{\mathrm{Var}(\xi)} \leq \Gamma, \quad \mathrm{Var}(\xi) \leq \Gamma_0 \right\}, \qquad \mathbf{\Gamma} \coloneqq [\Gamma_0, \Gamma]^{\top},$$

where *confounding strength* $\Gamma \geq 0$ determines our assumption on the variation in $\xi$ explained by $X$.

Assumption 3 adopts the widely used partial R-squared sensitivity model Cinelli & Hazlett (2019), itself a generalization of the classic Rosenbaum (2002). While we employ this model in our analyses, we do not necessarily restrict ourselves to it—under the linear Gaussian setting of Assumption 2, several families of PI and sensitivity models reduce to ellipsoidal constraints equivalent to the form:

**Lemma 2** (characterizing the identified set in a linear, Gaussian case)**.** *Under Assumptions 2 and 3,*

$$\mathcal{H}_{\mathrm{pi}} = \left\{ \boldsymbol{h} \;\middle|\; \|\boldsymbol{h} - \boldsymbol{h}_{\mathrm{erm}}\|_{\mathbf{\Sigma}_X}^2 \leq r(\mathbf{\Gamma})^2 \right\},$$

*where the ellipsoid radius $r(\mathbf{\Gamma}) \geq 0$ depends on the choice of constraint parameters. Furthermore,*

$$\mathcal{H}_{\mathrm{pi}}(\boldsymbol{x}) = \left[ \; \boldsymbol{h}_{\mathrm{erm}}^{\top}\boldsymbol{x} - r(\mathbf{\Gamma}) \cdot \|\boldsymbol{x}\|_{\mathbf{\Sigma}_X^{-1}}, \;\; \boldsymbol{h}_{\mathrm{erm}}^{\top}\boldsymbol{x} + r(\mathbf{\Gamma}) \cdot \|\boldsymbol{x}\|_{\mathbf{\Sigma}_X^{-1}} \; \right].$$

*Proof.* See Appendix A.5 for the proof. ☐

Our results thus carry broader implications for PI and sensitivity analysis, as we discuss in Section 7.

### 4.2.1 BETTER BOUNDS WITH DATA AUGMENTATION

First and foremost, we investigate if the post-DA bounds are, in some way, better than the baseline PI bounds. That is, if this exercise is useful at all. We present two results to support this claim.

**Proposition 2** (sharper bounds with DA)**.** *For Assumptions 1 to 3, Lebesgue measure (volume) $|\cdot|$,*

$$\frac{|\mathcal{H}_{\mathrm{da+pi}}|}{|\mathcal{H}_{\mathrm{pi}}|} = \sqrt{\frac{\det \mathbf{\Sigma}_X}{\det \mathbf{\Sigma}_{GX}}} < 1, \qquad \frac{|\mathcal{H}_{\mathrm{da+pi}}(\boldsymbol{x})|}{|\mathcal{H}_{\mathrm{pi}}(\boldsymbol{x})|} = \frac{\|\boldsymbol{x}\|_{\mathbf{\Sigma}_{GX}^{-1}}}{\|\boldsymbol{x}\|_{\mathbf{\Sigma}_X^{-1}}} \leq 1, \qquad \textit{equality iff} \qquad \boldsymbol{x} \perp \mathbf{\Delta}.$$

*Proof.* See Appendix A.4 for the proof. ☐

Proposition 2 states that the hypothesis set $\mathcal{H}_{\mathrm{da+pi}}$ is strictly smaller than the baseline $\mathcal{H}_{\mathrm{pi}}$. The same holds true for the intervals $\mathcal{H}_{\mathrm{da+pi}}(\boldsymbol{x})$ vs. $\mathcal{H}_{\mathrm{pi}}(\boldsymbol{x})$, unless the query point $\boldsymbol{x}$ is orthogonal to the variation induced by DA [1], in which case the size of the interval remains the same. Importantly, Proposition 2 shows that this increase in "sharpness" is a direct consequence of the added variation from of DA in Lemma 1. And because this variation is exogenous and independent of $\xi$, our hypothesis/assumption about the strength of confounding $\boldsymbol{\Gamma}$ in the system should remain the same. This combination of increased data variation, but same confounding assumptions is what reduces ambiguity in PI, resulting in the sharper bounds of Proposition 2. Lastly, in the "large DA" regime, the ellipsoid $\mathcal{H}_{\mathrm{da+pi}}$ collapses onto $\mathrm{null}(\boldsymbol{\Delta})$, and the interval width $|\mathcal{H}_{\mathrm{da+pi}}(\boldsymbol{x})|$ becomes $\left\|\boldsymbol{\Pi}_{\boldsymbol{\Delta}}^{\perp}\boldsymbol{x}\right\|_{\boldsymbol{\Sigma}_X^{-1}}^2$. Meaning, the DA removes *all but* the uncertainty that it cannot "see" within its $\mathrm{range}(\boldsymbol{\Delta})$.

Although smaller identified sets/ intervals are in general desirable, size alone may not be the most appropriate measure of "goodness" of the identified set. The next result is based on worst-case error.

**Theorem 1** (robust bounds with DA). *For $\mathcal{G}$-inv. $\boldsymbol{f}$, Assumptions 1 to 3, $\kappa := \lambda_{\max}\left(\boldsymbol{\Sigma}_X \boldsymbol{\Sigma}_{GX}^{-1}\right) \leq 1$,*

$$E_{\mathrm{worst}}^{\mathrm{do}(X)}(\mathcal{Q}_{\mathrm{pi}}) = \left(\|\boldsymbol{h}_{\mathrm{erm}} - \boldsymbol{f}\|_{\boldsymbol{\Sigma}_X} + r(\boldsymbol{\Gamma})\right)^2,$$

$$\overset{(i),(ii)}{\geq} \left(\underbrace{\|\boldsymbol{h}_{\mathrm{da+erm}} - \boldsymbol{f}\|_{\boldsymbol{\Sigma}_X}}_{\text{lower estimation error}} + \underbrace{\sqrt{\kappa} \cdot r(\boldsymbol{\Gamma})}_{\text{sharper bounds}}\right)^2 \overset{(ii)}{\geq} E_{\mathrm{worst}}^{\mathrm{do}(X)}(\mathcal{Q}_{\mathrm{da+pi}}).$$

*Equality iff (i) DA adds low variance $\kappa = 1$, and (ii) DA orthogonal to confounding $\boldsymbol{\Delta} \perp \boldsymbol{\Sigma}_{X,\xi}$. Also,*

$$\mathbb{E}_{\boldsymbol{x}}\left[E_{\mathrm{worst}}^{\mathrm{do}(\boldsymbol{x})}(\mathcal{Q}_{\mathrm{pi}})\right] > \underbrace{\|\boldsymbol{h}_{\mathrm{da+erm}} - \boldsymbol{f}\|_{\boldsymbol{\Sigma}_X}^2}_{\text{lower estimation error}} + \underbrace{\nu \cdot r(\boldsymbol{\Gamma})^2}_{\text{sharper bounds}} + s = \mathbb{E}_{\boldsymbol{x}}\left[E_{\mathrm{worst}}^{\mathrm{do}(\boldsymbol{x})}(\mathcal{Q}_{\mathrm{da+pi}})\right],$$

*where $\nu := \mathrm{tr}\left(\boldsymbol{\Sigma}_X \boldsymbol{\Sigma}_{GX}^{-1}\right) < \mathrm{tr}\left(\boldsymbol{\Sigma}_X \boldsymbol{\Sigma}_X^{-1}\right) = m$, queries $\boldsymbol{x} \sim \mathcal{N}(\mathbf{0}, \boldsymbol{\Sigma}_X)$ and some slack $s \geq 0$.*

*Proof.* See Appendix A.1 for the proof. $\qquad\square$

Theorem 1 shows that DA dominates PI on worst-case error through two mechanisms: (i) From Proposition 1, confounding aligned DA causes the PI centroid $\boldsymbol{h}_{\mathrm{erm}}$ to drift closer to $\boldsymbol{f}$, bringing the bounds with it, thereby reducing worst-case error. (ii) Independently, from Proposition 2, the bounds themselves shrink, pushing the worst-case point closer still to $\boldsymbol{f}$. Given that the worst-case error bounds how bad the performance of any one hypothesis in the identified set may be, application of a DA pre-processing to PI therefore makes subsequent decision making more robust and reliable. Theorem 1 also gives a lower bound on this improvement via the factor $\kappa$, which in the "large DA" regime approaches 0 when $\boldsymbol{\Delta}$ has full span on $\mathbb{R}^m$, but is 1 otherwise. In our linear setting of Assumption 2, the former implies a trivial $\boldsymbol{f}$, and so the last inequality in Theorem 1 more clearly shows the independent, and strictly positive (average) effect $\nu$ of sharper bounds for individual queries $\boldsymbol{x} \sim \mathbb{P}_X$. Which in the "large DA" regime shrinks to $\nu \to \dim(\mathrm{null}(\boldsymbol{\Delta})) =: k$, reducing by a factor $(m-k)/m < 1$, and directly improving (average) worst-case error for a random query $\boldsymbol{x}$.

### 4.2.2 VALID BOUNDS WITH DATA AUGMENTATION

Finally, we discuss perhaps the most important property in PI—bound validity. We address this as:

**Theorem 2** (valid bounds with DA). *For any $\mathcal{G}$-invariant $\boldsymbol{f}$, it holds under Assumptions 1 to 3 that*

$$E_{\mathrm{approx}}^{\mathrm{do}(X)}(\mathcal{Q}_{\mathrm{da+pi}}) \leq E_{\mathrm{approx}}^{\mathrm{do}(X)}(\mathcal{Q}_{\mathrm{pi}}), \qquad \text{equality iff} \qquad \mathbb{P} \in \mathcal{Q}_{\mathrm{pi}}, \qquad \text{or} \qquad \boldsymbol{\Delta} \perp \boldsymbol{\Sigma}_{X,\xi}.$$

*Proof.* See Appendix A.2 for the proof. $\qquad\square$

Meaning the identified set $\mathcal{H}_{\mathrm{da+pi}}$ is no farther from $\boldsymbol{f}$ compared to the original set $\mathcal{H}_{\mathrm{pi}}$, and is strictly closer to $\boldsymbol{f}$ so long as the DA induced variation aligns with confounding. Of course it follows that when $\mathcal{Q}_{\mathrm{pi}}$ contains the true joint distribution $\mathbb{P}$, then $\boldsymbol{f} \in \mathcal{H}_{\mathrm{pi}}$ and so we should also have $\boldsymbol{f} \in \mathcal{H}_{\mathrm{da+pi}}$. Instead of such a simple set inclusion criteria, we keep the more general approximation error framing of Theorem 2 because we also position DA as a tool for improved sensitivity analysis

---

[1] Intuitively, this would be like rotating an image $\boldsymbol{x}$ of a centered circle —the rotation leaves $\boldsymbol{x}$ unchanged.

where the constraint set may not necessarily be valid for some values of $\boldsymbol{\Gamma}$. Theorem 2 is then reassures that with $\mathcal{G}$-invariant $\boldsymbol{f}$, DA at the very least should not cause $\mathcal{H}_{\text{pi}}$ to drift away from $\boldsymbol{f}$.

Immediately following from Theorem 2, when $\mathbb{P} \in \mathcal{Q}_{\text{pi}}$, we also get $\boldsymbol{f}^\top \boldsymbol{x} \in \mathcal{H}_{\text{da+pi}}(\boldsymbol{x}), \mathcal{H}_{\text{pi}}(\boldsymbol{x})$. It is difficult, however, to show a similar result as Theorem 1 for the point-wise evaluation of $E_{\text{approx}}^{\text{do}(\boldsymbol{x})}(\mathcal{Q}_{\text{da+pi}})$ vs. $E_{\text{approx}}^{\text{do}(\boldsymbol{x})}(\mathcal{Q}_{\text{pi}})$ when $\mathbb{P} \notin \mathcal{Q}_{\text{pi}}$, as the approximation error non-trivially depends on the alignment of unknown confounding $\boldsymbol{\Sigma}_{X,\xi}$ with the query $\boldsymbol{x}$, and both can be arbitrary.

## 5 RELATED WORK

**PI and sensitivity analysis.** We give an account of related PI and sensitivity analysis literature in Section 2.3. Our work is largely orthogonal but complementary to this: we introduce a new source of constraints—symmetry knowledge—that is compatible and composes with existing PI frameworks.

**Symmetry and invariance in causal inference.** Invariance is fundamental to causality: causal mechanisms yield predictions invariant to interventions (Peters et al., 2016). Methods enforce such invariances using auxiliary variables for identification (Peters et al., 2016; Heinze-Deml et al., 2018; Arjovsky et al., 2019; Dance & Bloem-Reddy, 2024; Kilbertus et al., 2020; Singh et al., 2019; Zhang et al., 2023) or robust prediction (Rothenhäusler et al., 2021; Krueger et al., 2021; Christiansen et al., 2022). Akbar et al. (2025) leverage symmetry knowledge for robust prediction; whereas we address the orthogonal, but more principled problem of PI when point identification is infeasible.

**Counterfactual DA.** The literature on counterfactual DA has been the main focus of causal analysis of DA. These methods achieve robust predictors by synthesizing counterfactual examples (Ilse et al., 2021; Yuan et al., 2024; Feder et al., 2023; Pitis et al., 2022; Armengol Urpí et al., 2024; Mahajan et al., 2021; Aloui et al., 2023), but require restrictive assumptions: full SEMs (Yuan et al., 2024; Feder et al., 2023), specific auxiliary variables (Ilse et al., 2021; Feder et al., 2023; Mahajan et al., 2021; Aloui et al., 2023), or complete causal graphs (Pitis et al., 2022; Armengol Urpí et al., 2024). We, like Akbar et al. (2025), require a more accessible symmetry knowledge about the data.

## 6 EXPERIMENTS

We validate our frameworks in finite samples. We fix the augmented sample size to match the original to show that bound sharpening stems from symmetry constraints rather than variance reduction.

### 6.1 SIMULATION EXPERIMENT

We follow Akbar et al. (2025) to instantiate a simulation for the linear Gaussian SEM from Assumption 2. To do this, we first sample the SEM parameters—standard normal matrix $\boldsymbol{T} \in \mathbb{R}^{m \times m}$, and vectors $\boldsymbol{f}, \boldsymbol{e} \in \mathbb{R}^m$, keeping them fixed throughout the experiment. Then sample standard normal exogenous variables $(U, N_X, N_Y)$ and pass them through the following model to get observable $(X, Y)$ confounded by the unobserved $U$ as:

$$X := \boldsymbol{T}^\top U + 0.1 \cdot N_X, \quad Y := \boldsymbol{f}^\top X + \boldsymbol{e}^\top U + 0.1 \cdot N_Y.$$

Next, we construct a DA $G$ such that $\boldsymbol{f}$ respects $\mathcal{G}$-invariance. As with Akbar et al. (2025), we do this by first taking the SVD of $\boldsymbol{f}$,

$$\boldsymbol{f} = \begin{bmatrix} \boldsymbol{u} & \boldsymbol{U}_0 \end{bmatrix} \begin{bmatrix} \sigma & \boldsymbol{0}_{1 \times (m-1)} \\ \boldsymbol{0}_{(m-1) \times 1} & \boldsymbol{0}_{(m-1) \times (m-1)} \end{bmatrix} \begin{bmatrix} \boldsymbol{v}^\top \\ \boldsymbol{V}_0^\top \end{bmatrix}.$$

The matrix $\boldsymbol{V}_0$ represents the orthonormal basis of $\text{null}(\boldsymbol{f})$. We take $k$ of these rows to construct $\boldsymbol{A} \in \mathbb{R}^{k \times m}$ which defines $G$:

$$GX := X + a \cdot \boldsymbol{A}^\top G, \quad G \sim \mathcal{N}(\boldsymbol{0}_k, \boldsymbol{I}_k).$$

Therefore, by construction we have $\mathcal{G}$-invariance and therefore $\boldsymbol{f}^\top X = \boldsymbol{f}^\top GX$. Figure 2 provides an intuitive visualization

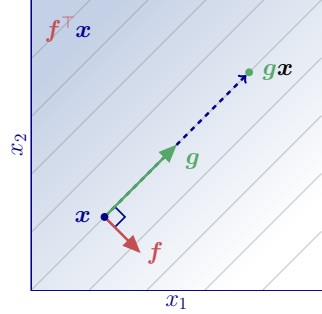

Figure 2: A cartoon visualization of the linear simulation setup. The augmentation generates $\boldsymbol{gx}$ by randomly translating $\boldsymbol{x}$ along the level-sets (contours) defined by the causal parameter $\boldsymbol{f}$, using additive noise sampled from the null-space of $\boldsymbol{f}$.

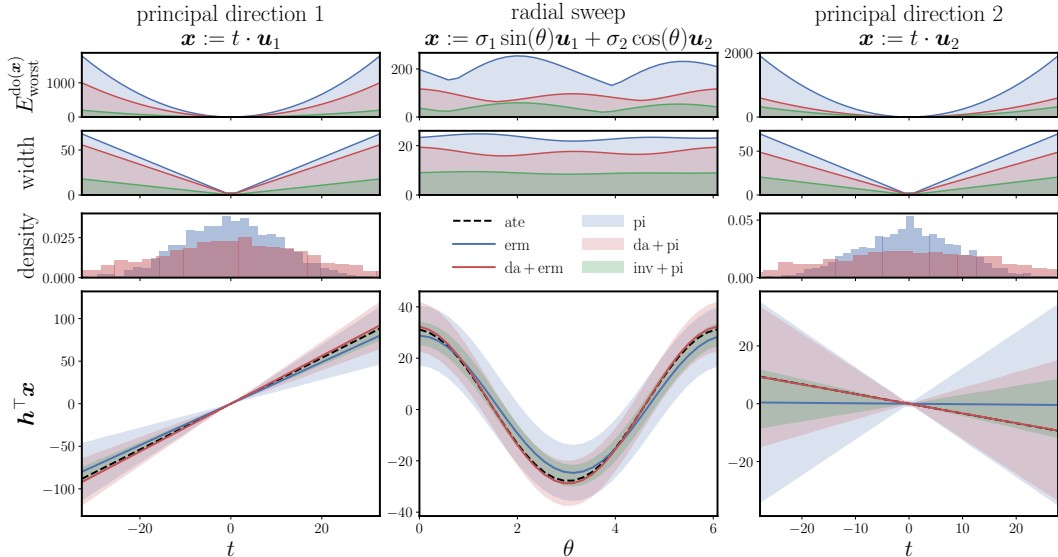

Figure 3: Data augmentation consistently sharpens partial identification bounds in a linear simulation. Across query points aligned with principal components (PC1, PC2) and a radial sweep, DA+PI (red) yields narrower intervals, lower worst-case excess risk ($E_{\mathrm{worst}}$), and predictions closer to the true average treatment effect (ATE, black dash) compared to baseline PI (blue).

of such a transformation. While this construction technically utilizes the ground truth $\boldsymbol{f}$, we treat access to $\boldsymbol{A}$ as representing prior domain knowledge about the functional symmetries, noting that this information alone is insufficient to identify $\boldsymbol{f}$ due to the unobserved confounding $\boldsymbol{e}^\top U$.

Taking $m = 32$, $k = 31$, we generate $n = 2048$ samples for $(X, Y, G)$ with the DA std parameter $a$ specified as 4. For ERM we use a closed form linear OLS solution. And for PI we use the partial R-squared sensitivity model from Assumption 3 for a range of query points $\boldsymbol{x}_0, \boldsymbol{x}_1$ with the sensitivity parameter set as $\Gamma = \Gamma_0 = 2^9$, and $\epsilon = 2^{-3}$ for the invariance error constraint.

To visualize the results, we chose $\boldsymbol{x} := t \cdot \boldsymbol{u}_1$ and $\boldsymbol{x} := t \cdot \boldsymbol{u}_2$ where $\boldsymbol{u}_1$, $\boldsymbol{u}_2$ are the first and second principal components of the data. We then sweep $t$ over $\pm 3$ standard-deviations, computing intervals $\mathcal{H}_{\mathrm{pi}}(\boldsymbol{x})$, $\mathcal{H}_{\mathrm{da+pi}}(\boldsymbol{x})$ via convex programming (separately for the upper and lower bounds). The results are shown in Fig. 3 (left, right). Fig. 3 (center) also shows a radial sweep over $\theta \in [0, 2\pi]$ to generate queries $\boldsymbol{x} := \sigma_0 \cdot \sin(\theta) \cdot \boldsymbol{u}_0 + \sigma_1 \cdot \cos(\theta) \cdot \boldsymbol{u}_1$.

## 6.2 Optical device experiment

We utilize the benchmark dataset provided by Janzing & Schölkopf (2018b), consisting of $3 \times 3$ pixel images $X$ displayed on a laptop screen which generate voltage readings $Y$ across a photodiode. The system involves a physically instantiated hidden confounder $U$ that controls the intensity of two LEDs; the first affects the webcam capturing $X$, while the second influences the photodiode measuring $Y$. We derive the ground-truth causal predictor $\boldsymbol{f}$ by regressing $Y$ on the joint features $(\phi(X), U)$, where $\phi(X)$ denotes polynomial features of $X$. We select the polynomial degree $d \in \{1, \cdots, 5\}$ that best explains the data (degree 2 in our case) and subsequently remove the learned component corresponding to $U$ to recover $\boldsymbol{f}$. Our choice of DA on $X$ includes additive Gaussian noise $G \sim \mathcal{N}(\boldsymbol{0}, \boldsymbol{\Sigma}_X/10)$, random vertical/horizontal flips and $90^0$ rotations for DA. We then compute the features $\phi(GX)$ to be used with PI, setting $\Gamma = \Gamma_0 = 10^2$ for the partial R-squared model from Assumption 3, and $\epsilon = 2^{-3}$ for the invariance error constraint on a datasets of $n = 1000$ samples. Figure 4 shows that DA+PI sharpens bounds over the PI baseline. The visualization approach in the same as in Section 6.1.

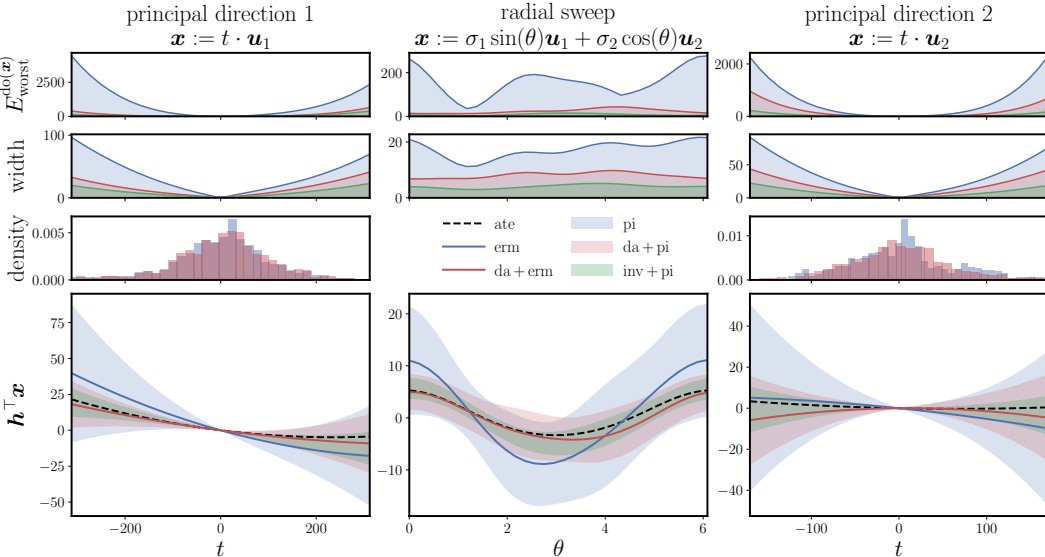

Figure 4: Our method sharpens causal bounds on the real-world Optical Device dataset. Even with complex, non-linear relationships, applying outcome-invariant DA (red) substantially narrows the partial identification bounds compared to the baseline (blue).

## 7 LIMITATIONS, ASSUMPTIONS AND FUTURE WORK

**Symmetry knowledge.** Our approach hinges on the untestable assumption that the target $f$ is $\mathcal{G}$-invariant for the chosen symmetry transformation. While this does require prior knowledge, our framework also allows to handle symmetry miss-specification in the explicit invariance error constraint from Section 4.1 via the $\epsilon$ parameter. Additionally, we remind the readers that untestable assumptions are fundamental for making any causal conclusions from observational data with unobserved confounding (Pearl, 2009), as is the norm in partial identification. This also includes access to auxiliary variables since the conditional independences that they represent are also merely untestable assumptions. Furthermore, Akbar et al. (2025) argues that a symmetry-based knowledge assumption is actually quite practical given its precedence in the DA and invariance literature (Chen et al., 2020; Lyle et al., 2020; Shao et al., 2022; Fawzi & Frossard, 2015; Dubois et al., 2021; Petrache & Trivedi, 2023; Montasser et al., 2024; Romero & Lohit, 2022; Zhu et al., 2021; Wong et al., 2016).

**Additional covariates.** Many works in PI and sensitivity analysis leverage access to additional auxiliary variables, such as instrumental variables (IVs) and observable confounders or back-doors (Kilbertus et al., 2020; Padh et al., 2023). Even though we do not explicitly model these to keep our analysis simple and tractable, we argue that our symmetry transformation framing is still compatible with them—for example, applying DA on $X$ does not invalidate an IV that enters into $X$.

**Additional partial identification approaches.** Many sensitivity and PI models can be reduced to the constraints in Assumption 3. These include, of course, the partial R-squared model, Rosenbaum (2002), MSM (under a mild bounded marginal ratio assumption), as well as DRO, Wasserstein, total-variation approaches. While a rigorous analysis is left for future work, it is important to specify that our results here are more general than just the partial R-squared model.

## 8 CONCLUSION

We show that causal symmetries sharpen partial identification bounds by restricting the hypothesis space. We operationalize this via explicit invariance constraints and implicit data augmentation. Through construction and linear analysis, respectively, we prove these methods yield valid, strictly tighter, and more robust bounds. Empirically validated and broadly compatible, our framework establishes symmetry as a powerful resource within the tool-belt for causal partial identification.

## ETHICS STATEMENT

The authors have read and adhered to the ICLR Code of Ethics. This work is primarily theoretical and methodological, focusing on the mathematical foundations of using data augmentation for partial identification. The experimental validation relies on a synthetic dataset generated for illustrative purposes and a standard, publicly available benchmark dataset (Optical Device). No human subjects were involved in this research, no new data was collected, and therefore, no Institutional Review Board (IRB) approval was required. The goal of this research is to improve the rigor and reliability of causal inference from observational data, which can lead to more robust and fair decision-making in various applications. We do not foresee any direct negative ethical implications or societal consequences stemming from this work.

## REPRODUCIBILITY STATEMENT

We are committed to ensuring the reproducibility of our work. All theoretical claims made in this paper are supported by detailed, step-by-step proofs, which can be found in the Appendix. The experimental setup for both the simulation study and the real-data experiment is described in Section 6. The complete source code to reproduce all experiments, figures, and results is included as supplementary material with this submission. The code is commented and contains all necessary implementation details, including hyperparameter settings and the specific data generation process for the simulation.

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

## A  PROOFS

### A.1  PROOF OF THEOREM 1—ROBUST BOUNDS WITH DA

**Theorem 1** (robust bounds with DA). *For $\mathcal{G}$-inv. $\boldsymbol{f}$, Assumptions 1 to 3, $\kappa := \lambda_{\max}\big(\boldsymbol{\Sigma}_X \boldsymbol{\Sigma}_{GX}^{-1}\big) \leq 1$,*

$$
E_{\mathrm{worst}}^{\mathrm{do}(X)}(\mathcal{Q}_{\mathrm{pi}}) = \left( \|\boldsymbol{h}_{\mathrm{erm}} - \boldsymbol{f}\|_{\boldsymbol{\Sigma}_X} + r(\boldsymbol{\Gamma}) \right)^2,
$$

$$
\overset{(i),\,(ii)}{\geq} \bigg( \underbrace{\|\boldsymbol{h}_{\mathrm{da+erm}} - \boldsymbol{f}\|_{\boldsymbol{\Sigma}_X}}_{\text{lower estimation error}} + \underbrace{\sqrt{\kappa} \cdot r(\boldsymbol{\Gamma})}_{\text{sharper bounds}} \bigg)^2 \overset{(ii)}{\geq} E_{\mathrm{worst}}^{\mathrm{do}(X)}(\mathcal{Q}_{\mathrm{da+pi}}).
$$

*Equality iff (i) DA adds low variance $\kappa = 1$, and (ii) DA orthogonal to confounding $\boldsymbol{\Delta} \perp \boldsymbol{\Sigma}_{X,\xi}$. Also,*

$$
\mathbb{E}_{\boldsymbol{x}}\left[ E_{\mathrm{worst}}^{\mathrm{do}(\boldsymbol{x})}(\mathcal{Q}_{\mathrm{pi}}) \right] > \underbrace{\|\boldsymbol{h}_{\mathrm{da+erm}} - \boldsymbol{f}\|_{\boldsymbol{\Sigma}_X}^2}_{\text{lower estimation error}} + \underbrace{\nu \cdot r(\boldsymbol{\Gamma})^2}_{\text{sharper bounds}} + s = \mathbb{E}_{\boldsymbol{x}}\left[ E_{\mathrm{worst}}^{\mathrm{do}(\boldsymbol{x})}(\mathcal{Q}_{\mathrm{da+pi}}) \right],
$$

*where $\nu := \mathrm{tr}\big(\boldsymbol{\Sigma}_X \boldsymbol{\Sigma}_{GX}^{-1}\big) < \mathrm{tr}\big(\boldsymbol{\Sigma}_X \boldsymbol{\Sigma}_X^{-1}\big) = m$, queries $\boldsymbol{x} \sim \mathcal{N}(\boldsymbol{0}, \boldsymbol{\Sigma}_X)$ and some slack $s \geq 0$.*

*Proof.* We show the two inequalities below in the respective sections.

**Population effect.**  Lemma 2, characterizes the identified sets $\mathcal{H}_{\mathrm{pi}}, \mathcal{H}_{\mathrm{da+pi}}$ as ellipsoids:

$$
\mathcal{H}_{\mathrm{pi}} = \left\{ \boldsymbol{h} \,\middle|\, \|\boldsymbol{h} - \boldsymbol{h}_{\mathrm{erm}}\|_{\boldsymbol{\Sigma}_X}^2 \leq r(\boldsymbol{\Gamma})^2 \right\}, \qquad \mathcal{H}_{\mathrm{da+pi}} = \left\{ \boldsymbol{h} \,\middle|\, \|\boldsymbol{h} - \boldsymbol{h}_{\mathrm{da+erm}}\|_{\boldsymbol{\Sigma}_{GX}}^2 \leq r(\boldsymbol{\Gamma})^2 \right\}.
$$

Now, from the definition of worst-case excess error in Section 2.3 it follows

$$
\begin{aligned}
E_{\mathrm{worst}}^{\mathrm{do}(X)}(\mathcal{Q}_{\mathrm{pi}}) &= \max_{\mathbb{Q} \in \mathcal{Q}_{\mathrm{pi}}} E^{\mathrm{do}(X)}\big(h_{\mathrm{adj}}^{\mathbb{Q}}\big), \\
&= \max_{\boldsymbol{h} \in \mathcal{H}_{\mathrm{pi}}} E^{\mathrm{do}(X)}(\boldsymbol{h}), && \text{(Re-parameterizing in terms of } \mathcal{H}_{\mathrm{pi}}.) \\
&= \max_{\boldsymbol{h} \in \mathcal{H}_{\mathrm{pi}}} \|\boldsymbol{h} - \boldsymbol{f}\|_{\boldsymbol{\Sigma}_X}^2, \\
&= \left( \|\boldsymbol{h}_{\mathrm{erm}} - \boldsymbol{f}\|_{\boldsymbol{\Sigma}_X} + r(\boldsymbol{\Gamma}) \right)^2, && \text{(Lemma 3)}
\end{aligned}
$$

where $r(\boldsymbol{\Gamma})$ is some constant entirely determined by $\boldsymbol{\Gamma}$. Now, we do a similar exercise with $\mathcal{Q}_{\mathrm{da+pi}}$,

$$
\begin{aligned}
&E_{\mathrm{worst}}^{\mathrm{do}(X)}(\mathcal{Q}_{\mathrm{da+pi}}) \\
&= \max_{\mathbb{Q} \in \mathcal{Q}_{\mathrm{da+pi}}} E^{\mathrm{do}(X)}\big(h_{\mathrm{adj}}^{\mathbb{Q}}\big), \\
&= \max_{\boldsymbol{h} \in \mathcal{H}_{\mathrm{da+pi}}} E^{\mathrm{do}(X)}(\boldsymbol{h}), && \text{(Re-parameterizing in terms of } \mathcal{H}_{\mathrm{da+pi}}.) \\
&= \max_{\boldsymbol{h} \in \mathcal{H}_{\mathrm{da+pi}}} \|\boldsymbol{h} - \boldsymbol{f}\|_{\boldsymbol{\Sigma}_X}^2, \\
&\overset{(\star)}{\leq} \boxed{\left( \|\boldsymbol{h}_{\mathrm{da+erm}} - \boldsymbol{f}\|_{\boldsymbol{\Sigma}_X} + r(\boldsymbol{\Gamma}) \cdot \sqrt{\lambda_{\max}\big(\boldsymbol{\Sigma}_X \boldsymbol{\Sigma}_{GX}^{-1}\big)} \right)^2,} \\
&&& \text{(Lemma 3, } = \text{iff } (\boldsymbol{h}_{\mathrm{da+erm}} - \boldsymbol{f}) \parallel \boldsymbol{v}_{\max}\big(\boldsymbol{\Sigma}_X \boldsymbol{\Sigma}_{GX}^{-1}\big).) \\
&\overset{(\dagger)}{\leq} \left( \|\boldsymbol{h}_{\mathrm{da+erm}} - \boldsymbol{f}\|_{\boldsymbol{\Sigma}_X} + r(\boldsymbol{\Gamma}) \right)^2, && \text{(Lemma 1, } \lambda_{\max}\big(\boldsymbol{\Sigma}_X \boldsymbol{\Sigma}_{GX}^{-1}\big) \leq 1.) \\
&\overset{(\ddagger)}{\leq} \left( \|\boldsymbol{h}_{\mathrm{erm}} - \boldsymbol{f}\|_{\boldsymbol{\Sigma}_X} + r(\boldsymbol{\Gamma}) \right)^2 = E_{\mathrm{worst}}^{\mathrm{do}(X)}(\mathcal{Q}_{\mathrm{pi}}). && \text{(Proposition 1, } = \text{iff } \boldsymbol{\Delta} \perp \boldsymbol{\Sigma}_{X,\xi}.)
\end{aligned}
$$

**Condition for equality.**  The bound involves three inequalities: $(\star)$ the geometric bound from Lemma 3, $(\dagger)$ the inflated variance implication from Lemma 1, and $(\ddagger)$ the estimation bound from Proposition 1. Of these, $(\dagger)$ is immediate, so we investigate $(\star), (\ddagger)$ in isolation with $\lambda_{\max} = 1$. Now, assuming further that the condition $\boldsymbol{\Delta} \perp \boldsymbol{\Sigma}_{X,\xi}$ is satisfied for $(\ddagger)$, it follows from Proposition 1 that

$$
\boldsymbol{h}_{\mathrm{da+erm}} = \boldsymbol{h}_{\mathrm{erm}} \qquad \Longleftrightarrow \qquad \boldsymbol{\Delta} \perp \boldsymbol{\Sigma}_{X,\xi}.
$$

So we go ahead and substitute $h_{\mathrm{da+erm}}$ with $h_{\mathrm{erm}}$ in $(\star)$. From Lemma 3, equality holds iff the bias vector $(h_{\mathrm{da+erm}} - f)$, now $(h_{\mathrm{erm}} - f)$, is a dominant eigenvector of $\Sigma_X \Sigma_{GX}^{-1}$. Because $(h_{\mathrm{erm}} - f) = \Sigma_X^{-1} \Sigma_{X,\xi}$ (OLS closed-form), and $\Sigma_{GX} = \Sigma_X + \Delta$ (Lemma 1), we follow the steps of Propositions 1 and 2 to do a change of basis by jointly diagonalizing $\Sigma_X, \Delta$ (Lemma 7) to show

$$\Sigma_X \Sigma_{GX}^{-1}(h_{\mathrm{erm}} - f) = \Sigma_X(\Sigma_X + \Delta)^{-1}\big(\Sigma_X^{-1}\Sigma_{X,\xi}\big) = (h_{\mathrm{erm}} - f), \quad \Longleftrightarrow \quad \Delta \perp \Sigma_{X,\xi}.$$

The bias $(h_{\mathrm{erm}} - f)$ is an eigenvector of $\Sigma_X \Sigma_{GX}^{-1}$ with eigenvalue $\lambda_{\max} = 1$. Therefore, equality holds for both $(\star)$ and $(\ddagger)$ iff $\Delta \perp \Sigma_{X,\xi}$ and the residual improvement is solely from radius contraction $\lambda_{\max} < 1$. Conditions for $(\star)$, $(\dagger)$, $(\ddagger)$ form conditions (i), (ii) in the statement.

**(Average) individual effect.** Define $J(\mathcal{Q}) := \mathbb{E}_x\Big[ E_{\mathrm{worst}}^{\mathrm{do}(x)}(\mathcal{Q}) \Big]$ for queries $x \sim \mathcal{N}(0, \Sigma_X)$. From Lemma 2, the worst-case risk at a query point $x$ is (bias + radius) squared:

$$E_{\mathrm{worst}}^{\mathrm{do}(x)}(\mathcal{Q}_{\mathrm{pi}}) = \left( \left| (h_{\mathrm{erm}} - f)^\top x \right| + r(\Gamma)\|x\|_{\Sigma_X^{-1}} \right)^2.$$

Expanding the square, we decompose the total expected risk for DA into three terms:

$$J(\mathcal{Q}_{\mathrm{da+pi}}) = \underbrace{\mathbb{E}_x\Big[ \|h_{\mathrm{da+erm}} - f\|_x^2 \Big]}_{\text{(a) estimation error}} + \underbrace{r(\Gamma)^2 \mathbb{E}_x\Big[ \|x\|_{\Sigma_{GX}^{-1}}^2 \Big]}_{\text{(b) average radius}},$$

$$+ \underbrace{2r(\Gamma)\mathbb{E}_x\Big[ \left|(h_{\mathrm{da+erm}} - f)^\top x\right| \|x\|_{\Sigma_{GX}^{-1}} \Big]}_{\text{(c) interaction of (a), (b)}}.$$

We analyze the reduction $J(\mathcal{Q}_{\mathrm{pi}}) - J(\mathcal{Q}_{\mathrm{da+pi}})$ term by term:

a) **Estimation error:** DA+ERM dominates ERM from Proposition 1, equality iff $\Delta \perp \Sigma_{X,\xi}$

b) **Average radius:** DA+PI *strictly* dominates PI from Proposition 2, as $x \not\perp \Delta$ almost surely. Also, expand $\|x\|_{\Sigma_{GX}^{-1}}^2$ into a trace term, and then use its cyclic permutation invariance gets

$$\mathbb{E}_x\Big[ \|x\|_{\Sigma_{GX}^{-1}}^2 \Big] = \mathrm{tr}\big(\Sigma_{GX}^{-1}\Sigma_X\big) < \mathrm{tr}\big(\Sigma_X^{-1}\Sigma_X\big) = m.$$

c) **Interaction term:** *Strictly* lower for any non-trivial DA $\Delta \neq 0$ from Lemma 5.

Concluding that:

$$J(\mathcal{Q}_{\mathrm{da+pi}}) < J(\mathcal{Q}_{\mathrm{pi}}) \quad \text{when} \quad \Delta \neq 0.$$

$\square$

## A.2 PROOF OF THEOREM 2—VALID BOUNDS WITH DA

**Theorem 2** (valid bounds with DA). *For any $\mathcal{G}$-invariant $\boldsymbol{f}$, it holds under Assumptions 1 to 3 that*

$$E_{\text{approx}}^{\text{do}(X)}(\mathcal{Q}_{\text{da+pi}}) \leq E_{\text{approx}}^{\text{do}(X)}(\mathcal{Q}_{\text{pi}}), \qquad \text{equality iff} \qquad \mathbb{P} \in \mathcal{Q}_{\text{pi}}, \qquad \text{or} \qquad \boldsymbol{\Delta} \perp \boldsymbol{\Sigma}_{X,\xi}.$$

*Proof.* From Lemma 2, we can characterize the identified sets $\mathcal{H}_{\text{pi}}$, $\mathcal{H}_{\text{da+pi}}$ as ellipsoids of the form

$$\mathcal{H}_{\text{pi}} = \left\{ \boldsymbol{h} \,\middle|\, \|\boldsymbol{h} - \boldsymbol{h}_{\text{erm}}\|_{\boldsymbol{\Sigma}_X}^2 \leq r(\boldsymbol{\Gamma})^2 \right\}, \qquad \mathcal{H}_{\text{da+pi}} = \left\{ \boldsymbol{h} \,\middle|\, \|\boldsymbol{h} - \boldsymbol{h}_{\text{da+erm}}\|_{\boldsymbol{\Sigma}_{GX}}^2 \leq r(\boldsymbol{\Gamma})^2 \right\}.$$

First consider $\mathbb{P} \notin \mathcal{Q}_{\text{pi}}$. Now, from the definition of approximation error in Section 2.3 it follows

$$\begin{aligned}
E_{\text{approx}}^{\text{do}(X)}(\mathcal{Q}_{\text{pi}}) &= \min_{\mathbb{Q} \in \mathcal{Q}_{\text{pi}}} E^{\text{do}(X)}\left(h_{\text{adj}}^{\mathbb{Q}}\right), \\
&= \min_{\boldsymbol{h} \in \mathcal{H}_{\text{pi}}} E^{\text{do}(X)}(\boldsymbol{h}), && \text{(Re-parameterizing in terms of } \mathcal{H}_{\text{pi}}.) \\
&= \min_{\boldsymbol{h} \in \mathcal{H}_{\text{pi}}} \|\boldsymbol{h} - \boldsymbol{f}\|_{\boldsymbol{\Sigma}_X}^2, \\
&= \left( \|\boldsymbol{h}_{\text{erm}} - \boldsymbol{f}\|_{\boldsymbol{\Sigma}_X} - r(\boldsymbol{\Gamma}) \right)^2, && \text{(Lemma 4)}
\end{aligned}$$

where $r(\boldsymbol{\Gamma})$ is some constant entirely determined by $\boldsymbol{\Gamma}$. Now, we do a similar exercise with $\mathcal{Q}_{\text{da+pi}}$,

$$\begin{aligned}
&E_{\text{approx}}^{\text{do}(X)}(\mathcal{Q}_{\text{da+pi}}) \\
&= \min_{\mathbb{Q} \in \mathcal{Q}_{\text{da+pi}}} E^{\text{do}(X)}\left(h_{\text{adj}}^{\mathbb{Q}}\right), \\
&= \min_{\boldsymbol{h} \in \mathcal{H}_{\text{da+pi}}} E^{\text{do}(X)}(\boldsymbol{h}), && \text{(Re-parameterizing in terms of } \mathcal{H}_{\text{da+pi}}.) \\
&= \min_{\boldsymbol{h} \in \mathcal{H}_{\text{da+pi}}} \|\boldsymbol{h} - \boldsymbol{f}\|_{\boldsymbol{\Sigma}_X}^2, \\
&\overset{(\heartsuit)}{\leq} \left( 1 - \frac{r(\boldsymbol{\Gamma})}{\|\boldsymbol{h}_{\text{da+erm}} - \boldsymbol{f}\|_{\boldsymbol{\Sigma}_{GX}}} \right)^2 \|\boldsymbol{h}_{\text{da+erm}} - \boldsymbol{f}\|_{\boldsymbol{\Sigma}_X}^2, \\
& && \text{(Lemma 4, } = \text{iff } (\boldsymbol{h}_{\text{da+erm}} - \boldsymbol{f}) \parallel \boldsymbol{v}(\boldsymbol{\Sigma}_X \boldsymbol{\Sigma}_{GX}^{-1}).) \\
&= \left( \|\boldsymbol{h}_{\text{da+erm}} - \boldsymbol{f}\|_{\boldsymbol{\Sigma}_{GX}} - r(\boldsymbol{\Gamma}) \right)^2 \frac{\|\boldsymbol{h}_{\text{da+erm}} - \boldsymbol{f}\|_{\boldsymbol{\Sigma}_X}^2}{\|\boldsymbol{h}_{\text{da+erm}} - \boldsymbol{f}\|_{\boldsymbol{\Sigma}_{GX}}^2}, \\
&\overset{(\spadesuit)}{\leq} \left( \|\boldsymbol{h}_{\text{erm}} - \boldsymbol{f}\|_{\boldsymbol{\Sigma}_X} - r(\boldsymbol{\Gamma}) \right)^2 = E_{\text{approx}}^{\text{do}(X)}(\mathcal{Q}_{\text{pi}}), && \text{(Similar to Proposition 1, } = \text{iff } \boldsymbol{\Delta} \perp \boldsymbol{\Sigma}_{X,\xi}.)
\end{aligned}$$

where the last inequality ($\ddagger$) follows from a similar approach as used in Proposition 1 to show that

$$\|\boldsymbol{h}_{\text{da+erm}} - \boldsymbol{f}\|_{\boldsymbol{\Sigma}_X}^2 \leq \|\boldsymbol{h}_{\text{da+erm}} - \boldsymbol{f}\|_{\boldsymbol{\Sigma}_{GX}}^2 \leq \|\boldsymbol{h}_{\text{erm}} - \boldsymbol{f}\|_{\boldsymbol{\Sigma}_X}^2,$$

which holds with equality if and only if $\boldsymbol{\Delta} \perp \boldsymbol{\Sigma}_{X,\xi}$. The case for $\mathbb{P} \in \mathcal{Q}_{\text{pi}}$ is trivial from Lemma 4.

**Condition for equality.** The two types of inequalities that comprise the given approximation error bound are Proposition 1-type estimation bias related ($\spadesuit$), and the ellipsoidal geometry inequality ($\heartsuit$) from Lemma 4. We can proceed similar to the corresponding section in Theorem 1 to show that:

$$\boldsymbol{\Delta} \perp \boldsymbol{\Sigma}_{X,\xi} \qquad \Longleftrightarrow \qquad (\boldsymbol{h}_{\text{da+erm}} - \boldsymbol{f}) \parallel \boldsymbol{v}_{\max}(\boldsymbol{\Sigma}_X \boldsymbol{\Sigma}_{GX}^{-1}).$$

Since condition of ($\heartsuit$) requires alignment of $(\boldsymbol{h}_{\text{da+erm}} - \boldsymbol{f})$ with *any* eigenvector $\boldsymbol{v}(\boldsymbol{\Sigma}_X \boldsymbol{\Sigma}_{GX}^{-1})$, therefore $\boldsymbol{v}_{\max}$ suffices. Consequently, equality holds for both ($\heartsuit$) and ($\spadesuit$) iff $\boldsymbol{\Delta} \perp \boldsymbol{\Sigma}_{X,\xi}$. $\qquad \square$

### A.3 Proof of Proposition 1—Estimation with DA (Akbar et al. (2025) lifted)

**Proposition 1** (estimation with DA (Akbar et al. (2025) lifted))**.** *For $\mathcal{G}$-inv. $\boldsymbol{f}$, Assumptions 1 and 2,*

$$0 \;\leq\; \frac{\kappa}{1+\kappa} \cdot \underbrace{\|\boldsymbol{\Pi}_{\boldsymbol{\Delta}}(\boldsymbol{h}_{\mathrm{erm}} - \boldsymbol{f})\|_{\boldsymbol{\Sigma}_X}^2}_{\text{estimation error within } \mathrm{range}(\boldsymbol{\Delta})} \;\leq\; E^{\mathrm{do}(X)}(\boldsymbol{h}_{\mathrm{erm}}) - E^{\mathrm{do}(X)}(\boldsymbol{h}_{\mathrm{da+erm}}),$$

$$\leq\; \|\boldsymbol{\Pi}_{\boldsymbol{\Delta}}(\boldsymbol{h}_{\mathrm{erm}} - \boldsymbol{f})\|_{\boldsymbol{\Sigma}_X}^2, \qquad \textit{eq. iff} \qquad \overbrace{\boldsymbol{\Delta} \perp \boldsymbol{\Sigma}_{X,\xi}}^{\substack{\textit{DA orthogonal} \\ \textit{to confounding}}},$$

*where $\kappa := \lambda_{\min}^+\left(\boldsymbol{\Sigma}_X^{-1}\boldsymbol{\Delta}\right) < \infty$ represents the lowest positive eigenvalue of the product $\boldsymbol{\Sigma}_X^{-1}\boldsymbol{\Delta}$.*

*Proof.* We start by first investigating the post-DA confounding vector $\mathbb{E}\left[(GX)\xi^\top\right] = \boldsymbol{\Sigma}_{GX,\xi}$ as

$$\begin{aligned} \boldsymbol{\Sigma}_{GX,\xi} &= \mathbb{E}\left[(GX)\xi^\top\right] \\ &= \mathbb{E}_{X,\xi}\left[\mathbb{E}_G[GX \mid X, \xi]\xi^\top\right] && \text{(Law of total expectation.)} \\ &= \mathbb{E}_{X,\xi}\left[\mathbb{E}_G[GX \mid X]\xi^\top\right] && (G \text{ exogenous} \implies G \perp\!\!\!\perp \xi \mid X.) \\ &= \mathbb{E}_{X,\xi}\left[X\xi^\top\right] = \boldsymbol{\Sigma}_{X,\xi}. && (\text{As } \mathbb{E}[GX \mid X] = X \text{ from Assumption 1.}) \end{aligned}$$

Now define $\boldsymbol{c} := \boldsymbol{\Sigma}_{X,\xi} = \boldsymbol{\Sigma}_{GX,\xi}$ for brevity. The estimation error in Eq. (6) for the baseline ERM and DA+ERM is governed by the projection of confounding $\boldsymbol{c}$ onto the respective data manifolds as:

$$E^{\mathrm{do}(X)}(\boldsymbol{h}_{\mathrm{erm}}) = \left\|\boldsymbol{\Sigma}_X^{-1}\boldsymbol{c}\right\|_{\boldsymbol{\Sigma}_X}^2 = \boldsymbol{c}^\top\boldsymbol{\Sigma}_X^{-1}\boldsymbol{c},$$

$$E^{\mathrm{do}(X)}(\boldsymbol{h}_{\mathrm{da+erm}}) = \left\|\boldsymbol{\Sigma}_{GX}^{-1}\boldsymbol{c}\right\|_{\boldsymbol{\Sigma}_X}^2 = \boldsymbol{c}^\top\boldsymbol{\Sigma}_{GX}^{-1}\boldsymbol{\Sigma}_X\boldsymbol{\Sigma}_{GX}^{-1}\boldsymbol{c}$$

Using $\boldsymbol{\Sigma}_X = \boldsymbol{\Sigma}_{GX} - \boldsymbol{\Delta}$ and the Resolvent Identity $\boldsymbol{\Sigma}_X^{-1} - \boldsymbol{\Sigma}_{GX}^{-1} = \boldsymbol{\Sigma}_{GX}^{-1}\boldsymbol{\Delta}\boldsymbol{\Sigma}_X^{-1}$, we get:

$$\begin{aligned} E^{\mathrm{do}(X)}(\boldsymbol{h}_{\mathrm{da+erm}}) &= \boldsymbol{c}^\top\boldsymbol{\Sigma}_{GX}^{-1}(\boldsymbol{\Sigma}_{GX} - \boldsymbol{\Delta})\boldsymbol{\Sigma}_{GX}^{-1}\boldsymbol{c} \\ &= \boldsymbol{c}^\top\boldsymbol{\Sigma}_{GX}^{-1}\boldsymbol{c} - \boldsymbol{c}^\top\left(\boldsymbol{\Sigma}_{GX}^{-1}\boldsymbol{\Delta}\boldsymbol{\Sigma}_{GX}^{-1}\right)\boldsymbol{c} \\ &= \left(\boldsymbol{c}^\top\boldsymbol{\Sigma}_X^{-1}\boldsymbol{c} - \boldsymbol{c}^\top\boldsymbol{\Sigma}_{GX}^{-1}\boldsymbol{\Delta}\boldsymbol{\Sigma}_X^{-1}\boldsymbol{c}\right) - \boldsymbol{c}^\top\left(\boldsymbol{\Sigma}_{GX}^{-1}\boldsymbol{\Delta}\boldsymbol{\Sigma}_{GX}^{-1}\right)\boldsymbol{c} \\ &= E^{\mathrm{do}(X)}(\boldsymbol{h}_{\mathrm{erm}}) - \boldsymbol{c}^\top\left(\boldsymbol{\Sigma}_{GX}^{-1}\boldsymbol{\Delta}\boldsymbol{\Sigma}_X^{-1}\right)\boldsymbol{c} - \boldsymbol{c}^\top\left(\boldsymbol{\Sigma}_{GX}^{-1}\boldsymbol{\Delta}\boldsymbol{\Sigma}_{GX}^{-1}\right)\boldsymbol{c}, \\ &= E^{\mathrm{do}(X)}(\boldsymbol{h}_{\mathrm{erm}}) \\ &\qquad - \overbrace{\boldsymbol{c}^\top\boldsymbol{\Sigma}_X^{-1}\left(\boldsymbol{\Sigma}_X\boldsymbol{\Sigma}_{GX}^{-1}\boldsymbol{\Delta}\right)\boldsymbol{\Sigma}_X^{-1}\boldsymbol{c}}^{0 \,\leq\, \text{first-order reduction}} \\ &\qquad\qquad - \underbrace{\boldsymbol{c}^\top\boldsymbol{\Sigma}_X^{-1}\left(\boldsymbol{\Sigma}_X\boldsymbol{\Sigma}_{GX}^{-1}\boldsymbol{\Delta}\boldsymbol{\Sigma}_{GX}^{-1}\boldsymbol{\Sigma}_X\right)\boldsymbol{\Sigma}_X^{-1}\boldsymbol{c}}_{0 \,\leq\, \text{second-order reduction}}. \end{aligned}$$

Both reduction terms are quadratic forms of the PSD matrix $\boldsymbol{\Delta}$ and are therefore non-negative.

Define $\delta$ as their sum. Lemma 6 lower-bounds the first-order term, and by extension lower-bounds $\delta$:

$$0 \;\leq\; \frac{\kappa}{1+\kappa} \cdot \|\boldsymbol{\Pi}_{\boldsymbol{\Delta}}(\boldsymbol{h}_{\mathrm{erm}} - \boldsymbol{f})\|_{\boldsymbol{\Sigma}_X}^2 \;\leq\; \text{first order term} \;\overset{(\blacktriangle)}{\leq}\; \delta.$$

Trace the same steps as Lemma 6 to bound $\delta$ from above via the simultaneous basis from Lemma 7 ($\boldsymbol{\Sigma}_X = \boldsymbol{S}^\top\boldsymbol{S}, \boldsymbol{\Delta} = \boldsymbol{S}^\top\boldsymbol{D}\boldsymbol{S}$). Taking $\boldsymbol{z} := \boldsymbol{S}\boldsymbol{\Sigma}_X^{-1}\boldsymbol{c}$ and eigenvalues $D_{ii}$ of $\boldsymbol{\Sigma}_X^{-1}\boldsymbol{\Delta}$, we can show

$$\delta \;=\; \sum_i z_i^2 \cdot \left(\overbrace{\underbrace{\frac{D_{ii}}{1+D_{ii}}}_{\text{1st order}} + \underbrace{\frac{D_{ii}}{(1+D_{ii})^2}}_{\text{2nd order}}}^{< 1}\right) \;\overset{(\nabla)}{\leq}\; \sum_{i:D_{ii}>0} 1 \cdot z_i^2 \;=\; \|\boldsymbol{\Pi}_{\boldsymbol{\Delta}}(\boldsymbol{h}_{\mathrm{erm}} - \boldsymbol{f})\|_{\boldsymbol{\Sigma}_X}^2.$$

**Condition for equality.** Equality holds for ($\blacktriangle$) iff $\boldsymbol{\Delta} \perp \boldsymbol{\Sigma}_{X,\xi}$, as otherwise the second-order term is strictly positive. Equality also holds for ($\nabla$) iff $\boldsymbol{\Delta} \perp \boldsymbol{\Sigma}_{X,\xi}$, because that entails $z_i = 0$ whenever $D_{ii} > 0$ so that the sums on both sides go to 0. $\qquad\square$

## A.4 PROOF OF PROPOSITION 2—SHARPER BOUNDS WITH DA

**Proposition 2** (sharper bounds with DA). *For Assumptions 1 to 3, Lebesgue measure (volume) $|\cdot|$,*

$$\frac{|\mathcal{H}_{\mathrm{da+pi}}|}{|\mathcal{H}_{\mathrm{pi}}|} = \sqrt{\frac{\det \boldsymbol{\Sigma}_X}{\det \boldsymbol{\Sigma}_{GX}}} < 1, \qquad \frac{|\mathcal{H}_{\mathrm{da+pi}}(\boldsymbol{x})|}{|\mathcal{H}_{\mathrm{pi}}(\boldsymbol{x})|} = \frac{\|\boldsymbol{x}\|_{\boldsymbol{\Sigma}_{GX}^{-1}}}{\|\boldsymbol{x}\|_{\boldsymbol{\Sigma}_X^{-1}}} \le 1, \qquad \textit{equality iff} \qquad \boldsymbol{x} \perp \boldsymbol{\Delta}.$$

*Proof.* We compare the geometric properties of the identified sets as characterized by Lemma 2.

**Ellipsoid volume (global contraction).** Given that the volume of a $\boldsymbol{\Sigma}$-ellipsoid $\propto (\det \boldsymbol{\Sigma})^{-1/2}$, it immediately follows from Lemmas 1 and 2, and the monotonicity of determinant for SPD matrices:

$$\boldsymbol{\Sigma}_X \preccurlyeq \boldsymbol{\Sigma}_{GX} \implies \det(\boldsymbol{\Sigma}_X) < \det(\boldsymbol{\Sigma}_{GX}),$$
$$\implies \det(\boldsymbol{\Sigma}_{GX})^{-1/2} < \det(\boldsymbol{\Sigma}_X)^{-1/2} \implies |\mathcal{H}_{\mathrm{da+pi}}| < |\mathcal{H}_{\mathrm{pi}}|.$$

**Interval width (point-wise contraction).** From Lemma 2, the width of the interval $\mathcal{H}_{\mathrm{pi}}(\boldsymbol{x})$ is simply $2r(\boldsymbol{\Gamma}) \cdot \|\boldsymbol{x}\|_{\boldsymbol{\Sigma}_X^{-1}}$. It then immediately follows from Lemma 1 and definition of the PSD order:

$$\boldsymbol{\Sigma}_X \preccurlyeq \boldsymbol{\Sigma}_{GX} \implies \boldsymbol{\Sigma}_{GX}^{-1} \preccurlyeq \boldsymbol{\Sigma}_X^{-1},$$
$$\implies \boldsymbol{x}^\top \boldsymbol{\Sigma}_{GX}^{-1} \boldsymbol{x} \le \boldsymbol{x}^\top \boldsymbol{\Sigma}_X^{-1} \boldsymbol{x} \implies |\mathcal{H}_{\mathrm{da+pi}}(\boldsymbol{x})| \le |\mathcal{H}_{\mathrm{pi}}(\boldsymbol{x})|.$$

**Condition for equality.** The interval width is strictly smaller for $\mathcal{H}_{\mathrm{da+pi}}(\boldsymbol{x})$ compared to $\mathcal{H}_{\mathrm{pi}}(\boldsymbol{x})$ unless the query point $\boldsymbol{x}$ lies in the null space of the difference $\boldsymbol{\Delta} := \boldsymbol{\Sigma}_{GX} - \boldsymbol{\Sigma}_X$. From Lemma 7,

$$\boldsymbol{\Sigma}_{GX}^{-1} = (\boldsymbol{\Sigma}_X + \boldsymbol{\Delta})^{-1} = (\boldsymbol{S}^\top \boldsymbol{S} + \boldsymbol{S}^\top \boldsymbol{D} \boldsymbol{S})^{-1} = \boldsymbol{S}^{-1} (\boldsymbol{I} + \boldsymbol{D})^{-1} \boldsymbol{S}^{-\top}.$$

When we analyze the ratio of squared norms using the basis $\boldsymbol{z} := \boldsymbol{S}^{-\top} \boldsymbol{x}$, it simplifies to:

$$\frac{\|\boldsymbol{x}\|_{\boldsymbol{\Sigma}_{GX}^{-1}}^2}{\|\boldsymbol{x}\|_{\boldsymbol{\Sigma}_X^{-1}}^2} = \frac{\boldsymbol{z}^\top (\boldsymbol{I} + \boldsymbol{D})^{-1} \boldsymbol{z}}{\boldsymbol{z}^\top \boldsymbol{z}} = \frac{\sum_i z_i^2 (1 + D_{ii})^{-1}}{\sum_i z_i^2}.$$

Since $\boldsymbol{D}$ is non-negative, the term $(1 + D_{ii})^{-1} < 1$ whenever $D_{ii} > 0$. Therefore, the ratio is strictly less than 1 unless $\boldsymbol{z}$ is supported only on indices where $D_{ii} = 0$. This requires $\boldsymbol{z}^\top \boldsymbol{D} \boldsymbol{z} = 0$, which transforms back to the condition that $\boldsymbol{x}$ must lie in the null-space of $\boldsymbol{\Delta}$ (i.e., $\boldsymbol{x} \perp \boldsymbol{\Delta}$). $\qquad\square$

## A.5 Miscellaneous supporting lemmas

**Lemma 1** (added exogenous variation with DA)**.** *Under Assumption 1, $G$ inflates the data variance,*

$$\boldsymbol{\Delta} := \boldsymbol{\Sigma}_{GX} - \boldsymbol{\Sigma}_X \succcurlyeq \mathbf{0}, \qquad \text{equality iff} \quad GX = X \quad a.s.$$

*Proof.* Represent $Z := GX$. Now, by applying the Law of Total Covariance conditioning on $X$,

$$\boldsymbol{\Sigma}_{GX} = \mathbb{E}[\,\mathrm{Cov}(GX\,|\,X)\,] + \mathrm{Cov}(\mathbb{E}[\,GX\,|\,X\,]). \tag{7}$$

By Assumption 1 (unbiased group action) we have $\mathbb{E}[\,GX\,|\,X\,] = X$, and the second term reduces to

$$\mathrm{Cov}(\mathbb{E}[\,GX\,|\,X\,]) = \mathrm{Cov}(X) = \boldsymbol{\Sigma}_X. \tag{8}$$

The first term represents the exogenous variation injected by the group action. Let $\boldsymbol{\Delta} = \mathbb{E}[\,\mathrm{Cov}(GX\,|\,X)\,]$. Since covariance matrices are PSD by definition, we have $\boldsymbol{\Delta} \succcurlyeq \mathbf{0}$.

**Condition for equality.** The inequality holds with equality ($\boldsymbol{\Sigma}_{GX} = \boldsymbol{\Sigma}_X$) iff the injected noise matrix $\boldsymbol{\Delta} = \mathbf{0}$. Since $\mathrm{Cov}(GX\,|\,X) \succcurlyeq \mathbf{0}$ almost surely, its expectation is zero if and only if $\mathrm{Cov}(GX\,|\,X) = \mathbf{0}$ almost surely. This implies $GX$ is a deterministic function of $X$. Given the unbiased assumption $\mathbb{E}[\,GX\,|\,X\,] = X$, this forces $GX = X$ almost surely (i.e., $G$ acts as identity over support of $X$). Therefore, for any non-trivial augmentation, the inequality $\boldsymbol{\Delta} \succcurlyeq \mathbf{0}$ is strict. $\square$

**Lemma 2** (characterizing the identified set in a linear, Gaussian case). *Under Assumptions 2 and 3,*

$$\mathcal{H}_{\mathrm{pi}} = \left\{ \boldsymbol{h} \;\middle|\; \left\| \boldsymbol{h} - \boldsymbol{h}_{\mathrm{erm}} \right\|_{\boldsymbol{\Sigma}_X}^2 \leq r(\boldsymbol{\Gamma})^2 \right\},$$

*where the ellipsoid radius $r(\boldsymbol{\Gamma}) \geq 0$ depends on the choice of constraint parameters. Furthermore,*

$$\mathcal{H}_{\mathrm{pi}}(\boldsymbol{x}) = \left[ \; \boldsymbol{h}_{\mathrm{erm}}^\top \boldsymbol{x} - r(\boldsymbol{\Gamma}) \cdot \|\boldsymbol{x}\|_{\boldsymbol{\Sigma}_X^{-1}}, \;\; \boldsymbol{h}_{\mathrm{erm}}^\top \boldsymbol{x} + r(\boldsymbol{\Gamma}) \cdot \|\boldsymbol{x}\|_{\boldsymbol{\Sigma}_X^{-1}} \; \right].$$

*Proof.* Compute the population covariance

$$\mathrm{Cov}(X, Y) = \mathrm{Cov}(X, \boldsymbol{f}^\top X + \xi) = \boldsymbol{\Sigma}_X \boldsymbol{f} + \boldsymbol{\Sigma}_{X,\xi},$$

so the (naïve) ERM estimand satisfies

$$\boldsymbol{h}_{\mathrm{erm}} = \boldsymbol{\Sigma}_X^{-1} \mathrm{Cov}(X, Y) = \boldsymbol{f} + \boldsymbol{\Sigma}_{XX}^{-1} \boldsymbol{\Sigma}_{X,\xi}.$$

Let $\boldsymbol{b} := \boldsymbol{h}_{\mathrm{erm}} - \boldsymbol{f} = \boldsymbol{\Sigma}_{XX}^{-1} \boldsymbol{\Sigma}_{X\xi}$. By the partial-$R^2$ constraint in Assumption 3

$$R_{\xi|X}^2 = \frac{\boldsymbol{\Sigma}_{X,\xi}^\top \boldsymbol{\Sigma}_X^{-1} \boldsymbol{\Sigma}_{X,\xi}}{\sigma_\xi^2} \leq \Gamma,$$

we have

$$\boldsymbol{\Sigma}_{X\xi}^\top \boldsymbol{\Sigma}_{XX}^{-1} \boldsymbol{\Sigma}_{X\xi} \leq \sigma_\xi^2 \Gamma.$$

Substituting $\boldsymbol{\Sigma}_{X\xi} = \boldsymbol{\Sigma}_{XX} \boldsymbol{b} = \boldsymbol{\Sigma}_{XX}(\boldsymbol{h}_{\mathrm{erm}} - \boldsymbol{f})$ yields

$$(\boldsymbol{h}_{\mathrm{erm}} - \boldsymbol{f})^\top \boldsymbol{\Sigma}_{XX} (\boldsymbol{h}_{\mathrm{erm}} - \boldsymbol{f}) \leq \sigma_\xi^2 \Gamma,$$

which is equivalent to

$$\|\boldsymbol{f} - \boldsymbol{h}_{\mathrm{erm}}\|_{\boldsymbol{\Sigma}_{XX}}^2 \leq \sigma_\xi^2 \Gamma \leq \Gamma_0 \Gamma.$$

Thus the identified set for $\boldsymbol{f}$ is the stated ellipsoid with radius $r(\boldsymbol{\Gamma})^2 = \Gamma_0 \Gamma$. The centred Gaussian assumption guarantees the linear projection interpretation used above is exact.

Lastly, since the identified set is an ellipsoid, maximizing/minimizing a linear functional $\boldsymbol{f}^\top \boldsymbol{x}$ is just moving along its principal axis in the direction of $\boldsymbol{x}$, giving us the bounds for $\mathcal{H}_{\mathrm{pi}}(\boldsymbol{x})$. $\qquad\square$

**Lemma 3** (upper bound on distance of a point to farthest point on ellipsoid). *Take ellipsoid $\mathcal{O} \subset \mathbb{R}^n$*

$$\mathcal{O} = \left\{ \boldsymbol{x} \,\middle|\, (\boldsymbol{x} - \boldsymbol{x}_0)^\top \boldsymbol{\Sigma}_0 (\boldsymbol{x} - \boldsymbol{x}_0) \leq r_0^2 \right\},$$

*with radius $r_0$, centered at $\boldsymbol{x}_0$ and shape defined by the SPD matrix $\boldsymbol{\Sigma}_0 \succ 0$. For some arbitrary point $\boldsymbol{y} \in \mathbb{R}^n$, denote its distance from the farthest point on $\mathcal{O}$ as weighted by an SPD $\boldsymbol{\Sigma} \succ 0$ with*

$$D_{\boldsymbol{\Sigma}}^{\max}(\boldsymbol{y}, \mathcal{O}) := \max_{\boldsymbol{x} \in \mathcal{O}} \|\boldsymbol{y} - \boldsymbol{x}\|_{\boldsymbol{\Sigma}}.$$

*This distance is upper bounded as follows, with $\boldsymbol{v}_{\max}$ as the eigenvector corresponding to $\lambda_{\max}$.*

$$D_{\boldsymbol{\Sigma}}^{\max}(\boldsymbol{y}, \mathcal{O}) \leq \|\boldsymbol{y} - \boldsymbol{x}_0\|_{\boldsymbol{\Sigma}} + r_0 \cdot \sqrt{\lambda_{\max}\big(\boldsymbol{\Sigma} \boldsymbol{\Sigma}_0^{-1}\big)},$$

$$\text{equality iff} \qquad \boldsymbol{y} - \boldsymbol{x}_0 \parallel \boldsymbol{v}_{\max}\big(\boldsymbol{\Sigma} \boldsymbol{\Sigma}_0^{-1}\big).$$

*Proof.* By triangle inequality

$$\|\boldsymbol{y} - \boldsymbol{x}\|_{\boldsymbol{\Sigma}} \leq \|\boldsymbol{y} - \boldsymbol{x}_0\|_{\boldsymbol{\Sigma}} + \|\boldsymbol{x}_0 - \boldsymbol{x}\|_{\boldsymbol{\Sigma}}.$$

Now, simply maximizing both sides over $\boldsymbol{x} \in \mathcal{O}$,

$$\max_{\boldsymbol{x} \in \mathcal{O}} \|\boldsymbol{y} - \boldsymbol{x}\|_{\boldsymbol{\Sigma}} \leq \max_{\boldsymbol{x} \in \mathcal{O}}(\|\boldsymbol{y} - \boldsymbol{x}_0\|_{\boldsymbol{\Sigma}} + \|\boldsymbol{x}_0 - \boldsymbol{x}\|_{\boldsymbol{\Sigma}}) = \|\boldsymbol{y} - \boldsymbol{x}_0\|_{\boldsymbol{\Sigma}} + \max_{\boldsymbol{x} \in \mathcal{O}} \|\boldsymbol{x}_0 - \boldsymbol{x}\|_{\boldsymbol{\Sigma}}.$$

The last term $\max_{\boldsymbol{x} \in \mathcal{O}} \|\boldsymbol{x}_0 - \boldsymbol{x}\|_{\boldsymbol{\Sigma}}$ is simply the radius of the ellipsoid in the $\boldsymbol{\Sigma}$–norm, which is equal to $r_0 \cdot \sqrt{\cdot \lambda_{\max}\big(\boldsymbol{\Sigma} \boldsymbol{\Sigma}_0^{-1}\big)}$. The result follows.

**Condition for equality.** The triangle inequality holds with equality iff $(\boldsymbol{y} - \boldsymbol{x}_0)$ and $(\boldsymbol{x} - \boldsymbol{x}_0)$ are collinear. The second term is maximized when $(\boldsymbol{x} - \boldsymbol{x}_0)$ aligns with the dominant eigenvector $\boldsymbol{v}_{\max}\big(\boldsymbol{\Sigma} \boldsymbol{\Sigma}_0^{-1}\big)$ (the generalized principal axis). Therefore, the total bound is tight iff $(\boldsymbol{y} - \boldsymbol{x}_0)$ is itself an eigenvector corresponding to $\lambda_{\max}(\boldsymbol{\Sigma} \boldsymbol{\Sigma}_0^{-1})$, i.e. $(\boldsymbol{y} - \boldsymbol{x}_0) \parallel \boldsymbol{v}_{\max}\big(\boldsymbol{\Sigma} \boldsymbol{\Sigma}_0^{-1}\big)$. $\qquad \square$

**Lemma 4** (upper bound on distance of a point to an ellipsoid). *Take the following ellipsoid $\mathcal{O} \subset \mathbb{R}^n$*

$$\mathcal{O} = \left\{ \boldsymbol{x} \,\middle|\, (\boldsymbol{x} - \boldsymbol{x}_0)^\top \boldsymbol{\Sigma}_0 (\boldsymbol{x} - \boldsymbol{x}_0) \le r_0^2 \right\},$$

*with radius $r_0$, centered at $\boldsymbol{x}_0$ and shape defined by the SPD matrix $\boldsymbol{\Sigma}_0 \succ 0$. For some arbitrary point $\boldsymbol{y} \in \mathbb{R}^n$, denote its distance from $\mathcal{O}$ as weighted by an SPD $\boldsymbol{\Sigma} \succ 0$ with the following notation*

$$D_{\boldsymbol{\Sigma}}^{\min}(\boldsymbol{y}, \mathcal{O}) := \min_{\boldsymbol{x} \in \mathcal{O}} \|\boldsymbol{y} - \boldsymbol{x}\|_{\boldsymbol{\Sigma}}.$$

*This distance is upper bounded by the following closed-form, with $\boldsymbol{v}(\cdot)$ as any arbitrary eigenvector.*

$$D_{\boldsymbol{\Sigma}}^{\min}(\boldsymbol{y}, \mathcal{O}) \le \begin{cases} 0, & \boldsymbol{y} \in \mathcal{O}, \\ \left(1 - \dfrac{r_0}{\|\boldsymbol{y} - \boldsymbol{x}_0\|_{\boldsymbol{\Sigma}_0}}\right) \|\boldsymbol{y} - \boldsymbol{x}_0\|_{\boldsymbol{\Sigma}}, & \boldsymbol{y} \notin \mathcal{O}, \end{cases}$$

$$equality\ iff \quad \boldsymbol{y} \in \mathcal{O}, \quad or \quad \boldsymbol{y} - \boldsymbol{x}_0 \parallel \boldsymbol{v}\!\left(\boldsymbol{\Sigma}\boldsymbol{\Sigma}_0^{-1}\right).$$

*Proof.* The result for $\boldsymbol{y} \in \mathcal{O}$ case is immediate. To show the bound for $\boldsymbol{y} \notin \mathcal{O}$, consider the ray

$$\boldsymbol{x}(r) := \boldsymbol{x}_0 + r \cdot (\boldsymbol{y} - \boldsymbol{x}_0), \quad r \in [0, 1],$$

going from the ellipsoid center $\boldsymbol{x}_0$ through $\boldsymbol{y}$. This ray intersects with the ellipsoid boundary at

$$r^* = \frac{r_0}{\|\boldsymbol{y} - \boldsymbol{x}_0\|_{\boldsymbol{\Sigma}_0}} \in (0, 1),$$

due to $\mathcal{O}$ being a sphere under a $\boldsymbol{\Sigma}_0$ weighted norm. The point $\boldsymbol{x}^* := \boldsymbol{x}(r^*)$ lies on the boundary.

$$\Rightarrow \boldsymbol{y} - \boldsymbol{x}^* = (1 - r^*) \cdot (\boldsymbol{y} - \boldsymbol{x}_0).$$

Since the closest point along an arbitrary ray is never closer than the true minimum, we have

$$\begin{aligned} D_{\boldsymbol{\Sigma}}^{\min}(\boldsymbol{y}, \mathcal{O}) &= \min_{\boldsymbol{x} \in \mathcal{O}} \|\boldsymbol{y} - \boldsymbol{x}\|_{\boldsymbol{\Sigma}}, \\ &\le \|\boldsymbol{y} - \boldsymbol{x}^*\|_{\boldsymbol{\Sigma}}, \\ &= (1 - r^*) \cdot \|\boldsymbol{y} - \boldsymbol{x}_0\|_{\boldsymbol{\Sigma}}, \\ &= \left(1 - \frac{r_0}{\|\boldsymbol{y} - \boldsymbol{x}_0\|_{\boldsymbol{\Sigma}_0}}\right) \|\boldsymbol{y} - \boldsymbol{x}_0\|_{\boldsymbol{\Sigma}}. \end{aligned}$$

**Condition for equality.** The condition for $\boldsymbol{y} \in \mathcal{O}$ case is trivial. For $\boldsymbol{y} \notin \mathcal{O}$, the minimum distance from $\boldsymbol{y}$ to the ellipsoid occurs at the boundary intersection of the ray $\boldsymbol{x}(r)$ iff the gradient of the *objective* $\boldsymbol{\Sigma}(\boldsymbol{y} - \boldsymbol{x})$ is parallel to the gradient of the *constraint* $\boldsymbol{\Sigma}_0(\boldsymbol{x} - \boldsymbol{x}_0)$ at the intersection point. Since $(\boldsymbol{x} - \boldsymbol{x}_0)$ is proportional to $(\boldsymbol{y} - \boldsymbol{x}_0)$ along the ray, this optimality condition requires:

$$\boldsymbol{\Sigma}(\boldsymbol{y} - \boldsymbol{x}_0) \propto \boldsymbol{\Sigma}_0(\boldsymbol{y} - \boldsymbol{x}_0) \quad \Longleftrightarrow \quad (\boldsymbol{y} - \boldsymbol{x}_0) \propto \boldsymbol{\Sigma}^{-1}\boldsymbol{\Sigma}_0(\boldsymbol{y} - \boldsymbol{x}_0).$$

Thus, the ray bound is exact if and only if $(\boldsymbol{y} - \boldsymbol{x}_0)$ is an (*any*) eigenvector of $\boldsymbol{\Sigma}^{-1}\boldsymbol{\Sigma}_0$. $\quad\square$

**Lemma 5** (centroid-radius interaction bound via coupling). *For $\boldsymbol{x} \sim \mathcal{N}(\boldsymbol{0}, \boldsymbol{\Sigma}_0)$, consider two constant vectors $\boldsymbol{b}_1, \boldsymbol{b}_2 \in \mathbb{R}^m$ (representing **centroid displacements**) and two symmetric positive definite matrices $\boldsymbol{\Sigma}_1, \boldsymbol{\Sigma}_2 \succ \boldsymbol{0}$ (representing respective **radius metrics**). Define the interaction integral:*

$$J(\boldsymbol{b}, \boldsymbol{\Sigma}) := \mathbb{E}_{\boldsymbol{x}}\Big[ |\boldsymbol{b}^\top \boldsymbol{x}| \cdot \sqrt{\boldsymbol{x}^\top \boldsymbol{\Sigma} \boldsymbol{x}} \Big].$$

*If $(\boldsymbol{b}_2, \boldsymbol{\Sigma}_2)$ has a strictly shorter "whitened" centroid and a strictly narrower radius (PSD-wise), i.e.,*

1. ***Centroid Contraction:*** $\|\boldsymbol{b}_2\|_{\boldsymbol{\Sigma}_0} < \|\boldsymbol{b}_1\|_{\boldsymbol{\Sigma}_0}$,

2. ***Radius Contraction:*** $\boldsymbol{\Sigma}_2 \prec \boldsymbol{\Sigma}_1$,

*then the interaction term strictly decreases:*

$$J(\boldsymbol{b}_2, \boldsymbol{\Sigma}_2) < J(\boldsymbol{b}_1, \boldsymbol{\Sigma}_1).$$

*Proof.* To evaluate the integral, we transform it into spherically symmetric coordinates (whitening).

**Whitening.** We can express the data vector $\boldsymbol{x}$ as a linear transformation of a standard normal vector $\boldsymbol{z} \sim \mathcal{N}(\boldsymbol{0}, \mathbf{I}_m)$ such that $\boldsymbol{x} = \boldsymbol{\Sigma}_0^{1/2} \boldsymbol{z}$. Substituting this into the centroid and radius terms:

$$\text{Centroid:} \quad |\boldsymbol{b}^\top \boldsymbol{x}| = |\boldsymbol{b}^\top \boldsymbol{\Sigma}_0^{1/2} \boldsymbol{z}| = |(\boldsymbol{\Sigma}_0^{1/2} \boldsymbol{b})^\top \boldsymbol{z}| = |\tilde{\boldsymbol{b}}^\top \boldsymbol{z}|,$$

$$\text{Radius:} \quad \sqrt{\boldsymbol{x}^\top \boldsymbol{\Sigma} \boldsymbol{x}} = \sqrt{\boldsymbol{z}^\top \boldsymbol{\Sigma}_0^{1/2} \boldsymbol{\Sigma} \boldsymbol{\Sigma}_0^{1/2} \boldsymbol{z}} = \sqrt{\boldsymbol{z}^\top \tilde{\boldsymbol{\Sigma}} \boldsymbol{z}},$$

where $\tilde{\boldsymbol{b}} := \boldsymbol{\Sigma}_0^{1/2} \boldsymbol{b}$ is the whitened centroid, and $\tilde{\boldsymbol{\Sigma}} := \boldsymbol{\Sigma}_0^{1/2} \boldsymbol{\Sigma} \boldsymbol{\Sigma}_0^{1/2}$ is the whitened radius metric.

**Rotational symmetry (coupling).** The expectation is now over the standard normal variable $\boldsymbol{z}$:

$$J = \mathbb{E}_{\boldsymbol{z}}\Big[ |\tilde{\boldsymbol{b}}^\top \boldsymbol{z}| \cdot \sqrt{\boldsymbol{z}^\top \tilde{\boldsymbol{\Sigma}} \boldsymbol{z}} \Big].$$

Since the distribution of $\boldsymbol{z}$ is spherically symmetric (invariant to rotations), the distribution of the dot product $\tilde{\boldsymbol{b}}^\top \boldsymbol{z}$ depends only on the length of $\tilde{\boldsymbol{b}}$. We can conceptually rotate the coordinate system for each scenario such that $\tilde{\boldsymbol{b}}$ aligns with the first basis vector $\boldsymbol{e}_1$. In this rotated frame, $|\tilde{\boldsymbol{b}}^\top \boldsymbol{z}| = \|\tilde{\boldsymbol{b}}\| \cdot |z_1|$. Crucially, note that $\|\tilde{\boldsymbol{b}}\| = \|\boldsymbol{\Sigma}_0^{1/2} \boldsymbol{b}\|_2 = \sqrt{\boldsymbol{b}^\top \boldsymbol{\Sigma}_0 \boldsymbol{b}} = \|\boldsymbol{b}\|_{\boldsymbol{\Sigma}_0}$. Thus:

$$J(\boldsymbol{b}, \boldsymbol{\Sigma}) = \|\boldsymbol{b}\|_{\boldsymbol{\Sigma}_0} \cdot \mathbb{E}_{\boldsymbol{z}}\Big[ |z_1| \cdot \sqrt{\boldsymbol{z}^\top \tilde{\boldsymbol{\Sigma}} \boldsymbol{z}} \Big].$$

**Comparison.** We now compare $J_1 = J(\boldsymbol{b}_1, \boldsymbol{\Sigma}_1)$ and $J_2 = J(\boldsymbol{b}_2, \boldsymbol{\Sigma}_2)$.

$$
\begin{aligned}
J_2 &= \|\boldsymbol{b}_2\|_{\boldsymbol{\Sigma}_0} \cdot \mathbb{E}_{\boldsymbol{z}}\Big[ |z_1| \cdot \sqrt{\boldsymbol{z}^\top \tilde{\boldsymbol{\Sigma}}_2 \boldsymbol{z}} \Big] \\
&< \|\boldsymbol{b}_1\|_{\boldsymbol{\Sigma}_0} \cdot \mathbb{E}_{\boldsymbol{z}}\Big[ |z_1| \cdot \sqrt{\boldsymbol{z}^\top \tilde{\boldsymbol{\Sigma}}_2 \boldsymbol{z}} \Big] && \text{(by centroid contraction)} \\
&< \|\boldsymbol{b}_1\|_{\boldsymbol{\Sigma}_0} \cdot \mathbb{E}_{\boldsymbol{z}}\Big[ |z_1| \cdot \sqrt{\boldsymbol{z}^\top \tilde{\boldsymbol{\Sigma}}_1 \boldsymbol{z}} \Big] && \text{(by radius contraction)} \\
&= J_1.
\end{aligned}
$$

The second inequality holds because $\boldsymbol{\Sigma}_2 \prec \boldsymbol{\Sigma}_1$ implies $\tilde{\boldsymbol{\Sigma}}_2 \prec \tilde{\boldsymbol{\Sigma}}_1$, so $\boldsymbol{z}^\top \tilde{\boldsymbol{\Sigma}}_2 \boldsymbol{z} < \boldsymbol{z}^\top \tilde{\boldsymbol{\Sigma}}_1 \boldsymbol{z}$ for all $\boldsymbol{z} \neq \boldsymbol{0}$. Since $|z_1|$ is non-negative and not always zero, the expectation strictly decreases. $\qquad \square$

**Lemma 6** (sandwich bounds for SPD-PSD weighted norms)**.** *For $n \times n$ matrices $\boldsymbol{A} \succ \boldsymbol{0}$, $\boldsymbol{B} \succcurlyeq \boldsymbol{0}$, denote the pseudo-inverse as $\boldsymbol{B}^{\dagger}$, and $\boldsymbol{\Pi_B} := \boldsymbol{B}^{\dagger}\boldsymbol{B}$ projects onto* $\mathrm{range}(\boldsymbol{B})$. *Then, for any $\boldsymbol{x} \in \mathbb{R}^n$,*

$$\underbrace{\frac{\kappa}{1+\kappa}}_{\text{shrinkage factor } \leq 1} \cdot \|\boldsymbol{\Pi_B x}\|_A^2 \quad \leq \quad \boldsymbol{x}^{\top}\boldsymbol{A}(\boldsymbol{A}+\boldsymbol{B})^{-1}\boldsymbol{B}\,\boldsymbol{x} \quad \leq \quad \|\boldsymbol{\Pi_B x}\|_A^2,$$

*for bounded minimum positive eigenvalue $\kappa := \lambda_{\min}^{+}\!\left(\boldsymbol{A}^{-1}\boldsymbol{B}\right) < \infty$. Equality holds for upper bound iff $\boldsymbol{x} \perp \boldsymbol{B}$, and lower bound iff $\boldsymbol{\Pi_B x}$ is entirely in eigen-space of $\boldsymbol{A}^{-1}\boldsymbol{B}$ corresponding to $\kappa$.*

*Proof.* From Lemma 7, we have $\boldsymbol{A} = \boldsymbol{S}^{\top}\boldsymbol{S}$ and $\boldsymbol{B} = \boldsymbol{S}^{\top}\boldsymbol{D}\boldsymbol{S}$ for invertible $\boldsymbol{S}$ and diagonal $\boldsymbol{D} \succcurlyeq \boldsymbol{0}$. Note that $\boldsymbol{D}$ are eigenvalues of $\boldsymbol{A}^{-1}\boldsymbol{B}$ by cyclic permutation invariance (i.e., $\lambda(\boldsymbol{AB}) = \lambda(\boldsymbol{BA})$).

Define the change of basis $\boldsymbol{z} := \boldsymbol{Sx}$. Then $\boldsymbol{x} = \boldsymbol{S}^{-1}\boldsymbol{z}$, and

$$\begin{aligned}
\boldsymbol{x}^{\top}\boldsymbol{A}(\boldsymbol{A}+\boldsymbol{B})^{-1}\boldsymbol{B}\,\boldsymbol{x} &= \boldsymbol{x}^{\top}\boldsymbol{S}^{\top}\boldsymbol{S}(\boldsymbol{S}^{\top}\boldsymbol{S}+\boldsymbol{S}^{\top}\boldsymbol{D}\boldsymbol{S})^{-1}\boldsymbol{S}^{\top}\boldsymbol{D}\boldsymbol{S}\,\boldsymbol{x} \\
&= \boldsymbol{x}^{\top}\boldsymbol{S}^{\top}\boldsymbol{S}(\boldsymbol{S}^{\top}(\boldsymbol{I}+\boldsymbol{D})\boldsymbol{S})^{-1}\boldsymbol{S}^{\top}\boldsymbol{D}\boldsymbol{S}\boldsymbol{x} \\
&= \boldsymbol{x}^{\top}\boldsymbol{S}^{\top}\boldsymbol{S}\boldsymbol{S}^{-1}(\boldsymbol{I}+\boldsymbol{D})^{-1}\boldsymbol{S}^{-\top}\boldsymbol{S}^{\top}\boldsymbol{D}\boldsymbol{S}\boldsymbol{x} \\
&= \boldsymbol{x}^{\top}\boldsymbol{S}^{\top}(\boldsymbol{I}+\boldsymbol{D})^{-1}\boldsymbol{D}\boldsymbol{S}\boldsymbol{x} \\
&= \boldsymbol{z}^{\top}(\boldsymbol{I}+\boldsymbol{D})^{-1}\boldsymbol{D}\,\boldsymbol{z} \\
&= \sum_i \frac{D_{ii}}{1+D_{ii}}z_i^2.
\end{aligned}$$

Similarly, for the projected norm, noting that $\boldsymbol{\Pi_B} = \boldsymbol{S}^{-1}\boldsymbol{D}^{\dagger}\boldsymbol{D}\boldsymbol{S}$ and $\|\boldsymbol{x}\|_A^2 = \|\boldsymbol{Sx}\|_2^2$:

$$\|\boldsymbol{\Pi_B x}\|_A^2 = \left\|\boldsymbol{S}\big(\boldsymbol{S}^{-1}\boldsymbol{D}^{\dagger}\boldsymbol{D}\boldsymbol{S}\big)\boldsymbol{x}\right\|_2^2 = \left\|\boldsymbol{D}^{\dagger}\boldsymbol{D}\boldsymbol{z}\right\|_2^2 = \sum_{i:D_{ii}>0} z_i^2.$$

**Upper bound.** Since $\frac{D_{ii}}{1+D_{ii}} < 1$ for all $D_{ii} > 0$, the following inequality is strict for any $z_i \neq 0$.

$$\boldsymbol{x}^{\top}\boldsymbol{A}(\boldsymbol{A}+\boldsymbol{B})^{-1}\boldsymbol{B}\,\boldsymbol{x} = \sum_i \frac{D_{ii}}{1+D_{ii}}z_i^2 \qquad \leq \qquad \sum_i z_i^2 = \|\boldsymbol{\Pi_B x}\|_A^2.$$

And equality holds iff $z_i = 0$ for *all* active indices, which implies $\boldsymbol{\Pi_B x} = \boldsymbol{0}$ (i.e., $\boldsymbol{x} \perp \boldsymbol{B}$).

**Lower Bound.** The function $f(d) = \frac{d}{1+d}$ is monotonically increasing for $d \geq 0$. Restricting our attention to the support of the vector (indices where $D_{ii} > 0$), we define $\kappa = \min\{D_{ii} : D_{ii} > 0\}$. It follows that for every active index, $\frac{D_{ii}}{1+D_{ii}} \geq \frac{\kappa}{1+\kappa}$. Summing over the support:

$$\sum_{i:D_{ii}>0} \frac{D_{ii}}{1+D_{ii}}z_i^2 \quad \geq \quad \sum_{i:D_{ii}>0} \frac{\kappa}{1+\kappa}z_i^2 \quad = \quad \frac{\kappa}{1+\kappa}\|\boldsymbol{\Pi_B x}\|_A^2.$$

For the inequality to become an equality, we require $\frac{D_{ii}}{1+D_{ii}} = \frac{\kappa}{1+\kappa}$ for every index $i$ where $z_i \neq 0$. This implies $D_{ii} = \kappa$ for all contributing dimensions. Geometrically, this means the vector $\boldsymbol{x}$ (after projection) must align only with the directions associated with the minimum eigenvalue $\kappa$. $\qquad\square$

**Lemma 7** (SPD, PSD joint denationalization via congruence Akbar et al. (2025)). *For any $n \times n$ matrices $A \succ 0$, $B \succeq 0$, there exists an $n \times n$ invertible $S$ and non-negative diagonal $D$ such that*

$$A = S^\top S, \qquad\qquad B = S^\top D S.$$

*Proof.* See (Akbar et al., 2025, Lem. 2), cf. (Horn & Johnson, 1985, Thm. 7.6.4, p. 465). □

## USE OF LARGE LANGUAGE MODELS

A large language model (LLM) was utilized as a writing assistant to help refine the prose, improve clarity, and ensure a consistent narrative tone during the preparation of this manuscript. The human authors directed this process, take full responsibility for the final content, and are solely responsible for all scientific contributions of this work.

