# OpenReview forum: "Causal Partial Identification with Data Augmentation"
_ICLR.cc/2026/Conference — Submitted to ICLR 2026_

### Official Review · Reviewer_2Gcy · 2025-10-20

**Soundness:** 3
**Presentation:** 1
**Contribution:** 2
**Rating:** 4
**Confidence:** 2

**Summary:**

The paper seeks to provide theoretical bounds for using data augmentation when coupled with partial identifications. The bounds proposed give both best- and worst-case scenarios for partially identifying the causal averages they employ, with respect to predefined causal risks.

**Strengths:**

Although I am no expert in neither domain augmentation nor partial identification, it seems to me that combining these two ideas is quite novel. The quality of writing is not incredibly good (see Weaknesses).

The theorems appear to be sound and (without taking a close look at the proofs) the conclusions do seem natural enough.

The concepts presented are clearly explained and it is clear that the theorems are algebraic derivations based on these definitions.

**Weaknesses:**

Adding an image to illustrate an observation of the dataset described in Section 6.2 would make the explanation more intelligible.

The abuse of notation in Section 3 (Line 202) was quite confusing at first. It seems to me that the notation employed could be formally introduced, to prevent confusions.

The first equation hints to a very general functional model, that is later replaced by a linear model. Possibly because the results of Vankadara et al., (2022) are only valid for linear models. Is this the only bottleneck that prevents extending the current framework to more general models?

### Typos and errors:
Several citations in the first pages (and possibly other parts of the text) should be citep, instead of citet. This makes reading of these pages quite painful and annoying.

- Line 20: ubiquitous.
- Line 273: extra 'e' in perturbs.
- Line 281: I suspect there are some parentheses or mathematical notation missing for $f^T x \in H_{da+pi}(x), H_{pi}(x)$.
- Line 461: out -> our.
- Line 464: identification.
- Line 466: Rosenbaum is inconsistently-cited (no year).
- Equation (5), the norms used are not clearly defined. The (very subtle) difference between $X$ and $\mathbf{x}$ inside the do operator, makes these hard to distinguish on a first read.

**Questions:**

The authors claim these bounds are valid for the infinite-data setting and naturally are only able to assess in a finite-data setting. However, it would be good to numerically assess how the derived bounds behave for different data-size regimes.

Outside of the optical device data used in Section 6, it is hard for me to envision the type of causal questions where the proposed tools would be useful. Could the authors point out other settings where this could be useful? Note that I am not requesting more experiments with this question, but they would be welcome.

In Theorems 1 and 2, I assume that the '_equality iff_' would also correspond to having slack equal to zero. Is this intuition correct? If so, how does the slack increase for different degrees of independence? Since the models are linear (and possibly Gaussian), measuring this dependence could be easily done by correlations/covariances.

On that note, the Gaussian assumption (over $G, U, N_X$, and $N_Y$) in Example 1 is quite strong (and it later affects all the theoretical developments), in what key ways does your contribution depend on this?

---

> ### Author Response · Authors · 2025-11-28
> **Response to Reviewer 2Gcy**
>
> We thank Reviewer `R. 2Gcy` for their constructive feedback and for recognizing the novelty of our approach. We appreciate the thorough list of corrections, which has helped us significantly improve the manuscript's readability.
>
> # **Weaknesses:**
>
> ## W1. Visualization of the Optical Device dataset.
>
> We appreciate the suggestion about communicating intuition via data visualization, however we would like to clarify that the input data consists of randomized 3x3 pixel patterns which, visually, appear as unstructured noise. Consequently, we felt that displaying the raw pixel inputs might not provide additional intuitive value.
>
> For the reviewer's reference, the physical data generation process (the "optical device" setup) is illustrated in detail on `page#19` of the original preprint, which can be viewed here:
>
> https://arxiv.org/pdf/1704.01430
>
> To ensure this setup is intelligible to future readers without requiring them to check the reference, we have revised the description in **`Sec. 6.2`**. The text now reads:
>
> *“We utilize the benchmark dataset provided by `Janzing & Schölkopf (2018b)`, consisting of* $3 \times 3$ *pixel images* $X$ *displayed on a laptop screen which generate voltage readings* $Y$ *across a photodiode. The system involves a physically instantiated hidden confounder* $U$ *that controls the intensity of two LEDs; the first affects the webcam capturing* $X$ *while the second influences the photodiode measuring* $Y$*.”*
>
> We hope this revision and the reference to the physical diagram clarify the nature of the dataset.
>
> ## W2. Notation in `Sec. 3`.
>
> Thank you for highlighting this. We have now abandoned the abused $\operatorname{do}(X\coloneqq GX)$ notation entirely, as we did not think it was adding much to begin with.
>
> We hope the revision has no such confusions, and remain open to other suggestions to remove ambiguities.
>
> ## W3. General functional model vs. Linear model bottleneck.
>
> We agree that the linearity assumption is a strong constraint; however, it is currently necessary for the results without introducing even stronger assumptions elsewhere in the framework.
>
> To address the reviewer’s concern about the "bottleneck":
>
> 1. **Stronger Assumptions:** While the results of `Vankadara et al., (2022)` are indeed derived for linear models, our primary bottleneck preventing an immediate extension to general models is the difficulty of characterizing the analog of `Lem. 1` for feature covariances in non-linear settings (e.g., when lifting the current results to an RKHS,) without using strong assumptions.
> 2. **Future Directions:** We *suspect* it may be possible to bypass this limitation by abandoning function spaces entirely and work directly with distributions, which is a promising direction for future work.
>
> **Clarifying claims on generality vs. linearity:**
> To ensure full transparency regarding these constraints, we have updated the paper as follows:
>
> - **Explicit Definition:** We now explicitly state the linear, Gaussian setting in the **Abstract (`line#23`)**, **Introduction (`line#76`)**, and in a dedicated **`Assumption 2` block in `Sec. 3`**.
> - **General Relevance:** We have introduced an explicit inv.-error constraint in **`Sec. 4.1`**. The generality of this method keeps the more general framing of our paper well justified.
>
> ### W4. Typos and errors:
>
> All fixed! Thanks for highlighting!
>
>
> # **Questions:**
>
> ## Q1. Numerical assessment of data-size regimes.
>
> While we generally agree with the reviewer that rigorous ablations, including sample size sweeps, are interesting for investigating finite-sample and consistency properties of methods, we would like to offer some clarification on our rationale for not doing so.
>
> - We deliberately fixed samples $n$ between baseline and augmented data to isolate the effect of *symmetry constraints* from *variance reduction*, the later being already well-understood for DA (being its main application) and therefore less interesting to investigate.
> - More importantly, by construction, the consistency of our DA+PI composition follows from the consistency of the baseline PI solver used—we believe such a sample-size-sweep experiment would therefore be more of a statement about the base PI solver than our DA pre-processing.
>
> Therefore, we believe, fixing samples $n$ is both more principled and honest evaluation approach.

---

> ### Author Response · Authors · 2025-11-28
> **Continued; Response to Reviewer 2Gcy**
>
> ## Q2. Applicability
>
> We appreciate this question and acknowledge that time constraints prevented additional experiments. We offer two perspectives on applicability:
>
> **1. Recognizing symmetries in established causal problems:**
>
> Identifying symmetries in causal mechanisms requires domain expertise—a fundamental necessity for causal reasoning from observational data. As methods researchers, we may lack sufficient domain knowledge to recognize such symmetries in every application. However, we point to promising directions:
>
> - **Molecular biology:** Drug response prediction with molecular symmetry groups. Many biological systems exhibit rotational, reflectional, or permutation symmetries (e.g., protein folding, molecular docking). See `Jumper et al., (2021, Nature)` on AlphaFold's use of SE(3)-equivariance, and `Townshend et al. (2021, Science)` on geometric deep learning for molecular property prediction.
> - **Economics:** Policy effects with exchangeable units. When analyzing treatment effects across similar economic units (regions, individuals, firms), *exchangeability* assumptions naturally encode symmetries. This connects to synthetic control methods (`Abadie et al., 2010, JASA`).
>
> **2. Recognizing causal problems where symmetries are established:**
>
> Alternatively, one could identify causal questions in domains where symmetries are already well-established: computer vision, NLP (`Ribeiro et al., 2020, ACL`), physics (Noether's theorem), and any domain where geometric deep learning applies (`Bronstein et al., 2021`).
>
> **But are these PI problems?** Many are! The robust ML literature addresses problems that fundamentally involve hidden confounding (`J. Peters et al., 2017`), and also recall that PI is fundamentally about robust decision-making under distributional uncertainty.
>
> **Concrete example—Colored MNIST `M. Arjovsky et al., 2019`:** This popular robust ML benchmark involves binary classification where the observed label $Y \in {0,1}$ is spuriously correlated with image color $\xi$ rather than the true digit class $\tilde{Y}$. The observed covariance $\Sigma_{X,Y}$ is dominated by confounding $\Sigma_{X,\xi}$, deviating from the true causal relationship $\Sigma_{X,\tilde{Y}}$. The true causal function is not identifiable. Our PI framework addresses this by:
>
> 1. Finding bounds (encoding assumptions about data generation, e.g., bounded $\Sigma_{X,\xi}$)
> 2. If lower bound > 0.5, classify as 1
> 3. If upper bound < 0.5, classify as 0
> 4. Otherwise, apply a tie-breaking decision rule per requirements
>
> One can then apply a battery of transformations to the images to see if bounds are sharpened—(small) rotation, translations, blur, hue, contrast, saturation, etc.
>
> In fact, we plan to include exactly such an experiment in future revisions!
>
> ## Better characterization of "Equality iff" and Slack.
>
> We agree with the reviewer on the two accounts they highlighted, and addressed them below:
>
> 1. **Better characterization of equality conditions:** The slack $s$ makes the equality iff conditions less clear. Our original handling was rather lazy by leaving it in there, so we investigated a bit more carefully to show the following, which really strengthens our results:
>     1. *Slack disappears in `Thm. 2`(*valid-bounds*) with eq. iff conditon:* But we do not quantify it since validity is the central concern and *not* improvement in approx.-error.
>     2. *Slack does not always disappear in `Thm. 1`(*robust-bounds*) with eq. iff condition:* This observation was quite insightful and really improved our narrative + results. We have updated `Thm. 1` accordingly to account for the part that does not automatically go to zero, giving us a very cool `Prop.1` (est.-error) vs. `Prop. 2` (sharper bounds) decomposition in the structure.
> 2. **Better characterization of degree of improvement**: Following your and `R. 1PwZ` ’s feedback, we made an effort to quantify the degree of improvement (from slack $s$ or otherwise) in our bounds whereever such a quantification might be useful. We did this by:
>     1. Explicitly quantifying improvement in bounds in `Prop.1` (est.-error) vs. `Prop. 2` (sharper bounds).
>     2. Writing `Thm. 1` (*robust-bounds*) as a decomposition: *est.-error* (`Prop.1`) + *sharpness measure* (\~ `Prop. 2`), quantification of improvement **cascades** from them. We do not do this for `Thm. 2`(*valid-bounds*) as approx.-error improvement is not relevant for validity.
>
> We believe these changes should address the reviewers question on slack $s$ investigation.

---

> > ### Author Response · Authors · 2025-11-28
> > **Continued; Response to Reviewer 2Gc**
> >
> > ## Q4. Dependence on the Gaussian Assumption.
> >
> > The key utility of the Gaussianity for our analysis is that they reduce to **ellipsoidal constraints** (`Lem. 2`, now moved to main text), which makes it tractable to quantify bounds on *approx.-error* (`Thm. 2`) and *worst-error* (`Thm. 1`) as distances to said ellipsoids (`Lem. 3, 4`). In addition to these quadratic (ellipsoidal) constraints, our results can also be shown with additional linear/polytope constraints (e.g., “positivity” constraints are quite common in PI literature), although the proofs get hairy with many “iff” conditions, and readability takes a hit.
> >
> > We believe an extension of this to general finite second-moment distributions is not so trivial since they yield arbitrary convex constraint sets, making it difficult to compute bounds for *approx.-error and worst-error*. Unless, perhaps, one changes the tools of investigation (e.g., *information theoretic* metrics instead of our *risk-based* ones), but I have not explored these directions enough to confidently comment on them just yet, and leave them for future work.
> >
> > That being said, we believe the invariance error constraint in `Sec. 4.1` should alleviate some concerns regarding the usefulness of our framework for general, non-Gaussian settings.
> >
> > # Concluding
> >
> > We once again thank the reviewer for the inquisitive questions and feedback, addressing which has meaningfully improved the paper.

---

### Official Review · Reviewer_dc4P · 2025-10-28

**Soundness:** 2
**Presentation:** 3
**Contribution:** 2
**Rating:** 4
**Confidence:** 3

**Summary:**

The authors investigated whether outcome‑invariant data augmentation (DA) can sharpen partial identification (PI) bounds for causal effects under the presence of unmeasured confounders. The key idea lies in constructing an outcome-invariant data augmentation transformations G as a transformation intervention do(x=Gx). A very simple linear Gaussian case study is provided. By experiments, the authors showed that DA could help reduce the length for the pointwise interval width. For the theoretical analysis, the authors showed that DA lowers worst‑case causal excess risk over the identified set. This seems to be a flexible approach as the DA approach is model‑agnostic and compatible with many PI frameworks.

**Strengths:**

1.	The conceptual idea is very clear and straightforward. Constructing a outcome-invariant DA as transformation intervation and then use it as a plug-in module in any PI pipeline.
2.	The theoretical analysis is solid, especially for the and lower worst case excess risk (Theorem 2)
3.	The proposal is a pre‑processing step that can be composed with existing PI methods without changing their solvers or constraint sets.

**Weaknesses:**

1.	The linear–Gaussian SEM seems still restrictive with additive noise and a particular sensitivity model. Under other common cases, such as non-Gaussian, non-linear outcome model, many guarantees may not work.
2.	The outcome-invariant assumption is hard to be testable, there is no diagnostic or stress test for misspecified augmentations.
3.	The numerical results are not comprehensive enough. Only one real dataset (Optical Device) is used with fixed sample size (n=1000).

**Questions:**

1.	on the Optical Device data, why should flips/rotations/Gaussian noise be outcome‑invariant for f Can you provide predictive‑invariance checks?
2.	Can any of the theoretical results be extended to non-Gaussian case? Or with some additional mild assumption?
3.	How to choose G seems very important and tricky. A practical guideline for choosing G is very necessary.
4.	A comprehensive sensitivity analysis to mis‑specified DA is very important for understanding the role of DA.
5.	There are many augmentation parameters, such as noise scale, angle, etc. How these parameters affect the PI and the prediction performance?
6.	The authors should test DA in multiple PI framework to demonstrate compatibility and gains beyond the partial R^2 model.
7.	Since authors argue compatibility with IVs, a toy example where an IV is present, showing that DA preserves IV validity and may further tightens bounds, should be very convincing.

---

> ### Author Response · Authors · 2025-11-28
> **Response to Reviewer dc4P**
>
> We thank reviewer `R. dc4P` for their feedback and for recognizing our approach's flexibility and theoretical solidity.
>
> # Weaknesses
>
> ## W1. Linear-Gaussian SEM:
>
> **Fix#1; Generality:** We *partially* addressed implications of our framework to general settings by:
>
> - **More general DA:** Making our DA construction more general, only requirement is “identity centred” DA (`Assumption 1`, `Sec. 2.4`).
> - **Other PI models:** We clarify in `line#301` that our use of the particular partial-R^2 sensitivity model does not necessarily mean the results are not applicable to other PI approaches, since several standard PI models reduce to ellipsoids constraints of `Lem. 2` (now moved to main text) under linear, Gaussian setting. The same is also emphasized in `line#313`, `line#528`.
> - **Explicit invariance-constraint:** Proposing and discussing the *inv.-error* constraint approach in `Sec. 4.1`: Compatible with general, non-linear, Monte Carlo, gradient-based PI methods. We also comment on its behaviour with $\epsilon = 0$ in linear setting in `Sec. 4.2` via the “large DA” regime results that actually follow from simple DA.
>
> **Fix#2; Explicit, early communication of limitations:** Nevertheless, we recognise the limitation of our theoretical results and have stated the linear Gaussian assumption explicit in the **(1)** abstract `line#23`, **(2)** introduction `line#77`, and **(3)** dedicated `Assumption 2` block in `Section 3`.
>
> ## W2. Testability of $\mathcal{G}$-invariance
>
> $\mathcal{G}$-invariance is an *untestable assumption*—but this is unavoidable for any causal inference from observational data with unobserved confounding (`Pearl, 2009`). Consider that:
>
> 1. **All PI methods require untestable assumptions**: IV validity, partial R² bounds, propensity score sensitivity parameters—none can be verified from observational data alone
> 2. **Practical accessibility**: Symmetry assumptions are pervasive and well-established in ML literature (`Chen et al., 2020`; `Lyle et al., 2020`; `Shao et al., 2022`; and others in `Sec. 7`). Domain experts routinely identify such symmetries for DA—we propose incorporating these existing, established symmetries into PI.
>
> **Miss-specification Fix (`Sec. 4.1`):** Nevertheless, we agree that miss-specification may be an issue, and we *partially* address this concern in `Sec. 4.1` by introducing an *inv.-error constraint* for PI. This provides us a “knob” $\epsilon$ we can use to define belief in the amount of DA miss-specification, making our framework relevant even for miss-specified symmetry.
>
> ## W3. Additional Experiments
>
> We acknowledge the experimental section can be improved. Our motivation with and design of the presented experiments was to demonstrate:
>
> - ✓ Theoretical results hold in finite samples
> - ✓ Method works on both synthetic (full control) and real data (optical device)
> - ✓ Consistent improvement across different query points (PC1, PC2, radial sweep)
> - ✓ All three metrics improve: worst-error, interval width, validity (no invalid bound)
>
> Nevertheless, we agree that the experiments can be improved. For now, we have provided **additional results for the *inv.-error* constraint**. Unfortunately, time did not permit to conduct further experiments at this moment, but we shall consider doing so for future submissions.

---

> ### Author Response · Authors · 2025-11-28
> **Continued; Response to Reviewer dc4P**
>
> # Questions
>
> ## Q1. Justification for choice of DA in Optical Device
>
> The justification simply comes from intuition (i.e. *prior knowledge* about the task)—the task is fundamentally a photo-diode taking reading of a 3x3 image. It stands to reason then if you rotate the image, it should *not* change the photo-diode reading. The orientation of an image on a screen *should not* affect the irradiance (intensity of radiation) recorded from the image. The reasoning for Gaussian noise is less principled/justified, mostly relying on the invariance holding *approximately* as we add a *small* noise.
>
> We have not conducted invariance checks per-se, and even eye-balled most sensitivity parameters for PI (`Sec. 4.1`). But the idea is compelling to explore in future iterations.
>
> **NOTE:** Even with miss-specified transformations, the inv.-constraint PI approach (`Sec. 4.1`) should dominate baseline approaches so long as amount of miss-specification $\epsilon$ is well specified.
>
> ## Q2. Non-Gaussian extension
>
> In this draft, no. The main utility of the Gaussianity is ellipsoidal bounds (`Lem. 2`), which gives us useful and quantifiable bounds for *approx.-error* and *worst-error* via distances to ellipsoids. We believe the fundamental principles of PI improvement (`Lem. 1`) *should **conceptually** carry over*, but the tractability will be *significantly* more difficult for general finite second-moments. Unless, perhaps, one changes the tools of investigation (e.g., *distribution* sets instead of function sets, and *information theoretic* metrics instead of *risk-based* ones), but I have not explored these enough to confidently comment on them just yet, and leave them for future work.
>
> Nevertheless, we believe the inv.-error constraint in `Sec. 4.1` should alleviate some concerns regarding the usefulness of our framework for general, non-Gaussian settings.
>
> ## Q3. Practical guide for the choice of G
>
> As with all causal inference methods addressing hidden confounding: **domain knowledge is essential**. The choice of $G$ should be guided by reasoning about the causal mechanism—similar to justifying a candidate IV or sensitivity parameters.
>
> Interestingly, such symmetries and data augmentation are already known and established in certain applications (computer vision, NLP, physics, molecular biology, see our response to `R. 2Gcy` ’s `Question 2`). So rather than finding new symmetries from scratch, practitioners can often **repurpose** existing ones for the new task of PI. And should they suspect miss-specification, the approach in `Sec. 4.1` shows how they can still benefit from access to approximate symmetries. This makes the framework more accessible than it might initially appear.
>
> ## Q4+Q5. Sensitivity to DA miss-specification and parameters
>
> We agree that a comprehensive sensitivity analysis would be valuable. While the scope of the current work (establishing the fundamental theory) prevented a full empirical sweep, the **Invariance Error Constraint (`Sec. 4.1`)** provides the formal mechanism to do this. By varying $\epsilon$, one can rigorously test how sensitive the bounds are to violations of the symmetry assumption. We view this as a promising direction for future applied work.
>
> ## Q6+Q7. Compatibility with other PI models, including IVs
>
> We again agree these experiments would strengthen the empirical contribution. Time constraints prevented us from implementing them for this submission, but we emphasize that **conceptually, there is little reason to think our work does not extend to IVs and other standard PI models**.
>
> **Justification:**
>
> - **Other PI models:** As we clarify in `Sec. 4.2` , `line#301` that under the linear, Gaussian setting many PI and sensitivity models reduce to *ellipsoidal constraints* of `Lem. 2`. Therefore similar improvements should follow by analogous reasoning.
> - **IV based PI:** For a variable $Z$ to be a valid IV, it needs: (i) unconfoundedness $Z\perp\\!\\!\\!\perp \xi$, (ii) exclusion restriction $Y\perp\\!\\!\\!\perp Z | X$. Both properties continue to be satisfied post-DA:
>    - Unconfoundedness: $Z\perp\\!\\!\\!\perp \xi$ (unchanged, as DA doesn't affect $Z$ or $\xi$)
>    - Exclusion restriction: $Y\perp\\!\\!\\!\\perp Z | GX$ (preserved under transformation)
> - The *inv.-error constraint* (`Sec. 4.1`) is explicitly designed to be model-agnostic and compatible with any PI framework that accepts constraints on the data generating process.
>
> We plan to demonstrate compatibility across multiple PI frameworks (IVs, front-door criterion, proximal inference, bounds with multiple sensitivity parameters) with concrete toy examples in future work.
>
> # Closing
>
> We would like to thank reviewer `R. dc4P` once again for their remarks, and hope to incorporate stronger empirical results in future revisions.

---

### Official Review · Reviewer_ba8S · 2025-10-31

**Soundness:** 4
**Presentation:** 1
**Contribution:** 2
**Rating:** 2
**Confidence:** 3

**Summary:**

This paper proposes a method based on data augmentation to improve partial identification bounds in causal inference when unobserved confounding prevents point identification. The key idea is to use data augmentation, specifically, an outcome-invariant augmentation, as an auxiliary source of information that can act as a ``soft intervention.''
If we can transform data in ways that leave the outcome function unchanged (say, rotations of a picture in a classification task), these transformations can help tighten the partial id bounds.
The method requires background knowledge about the regression function, for instance the null space of the coefficients matrix in a linear regression model, which can be quite limiting in practice.

**Strengths:**

1. Even though the analysis is restricted to linear-Gaussian systems, I believe most of the results have straightforward generalizations to more complex models.

---
2. Synthetic and semi-synthetic experiments consistently show that DA+PI yields tighter bounds than baseline PI.

**Weaknesses:**

1. Significant overlap with Akbar et al [1]:
Taking a look at the first reference of the paper, one can notice immediately that this paper has a significant overlap with, and is essentially not adding much to Akbar et al. Surprisingly, a great deal of content is copied without even rewording. Ideas are copied from that paper: DA as soft intervention, Figure 2 of the paper with its caption, the running example of the paper, and most significantly, the theoretical results of this paper such as proposition 2, Lemma 3 and Lemma 4, all appear in Akbar et. al., and the rest of the results provided here are quite trivial or straightforward at best to prove given those.
Compared to Akbar et al., I don't see much of a novel idea, novel proof, novel result, or novel presentation, begging the question what is the merit of this paper given that? Only bringing up the observation that DA can be used for partial identification too? Then I do not believe this paper is contributing enough to the literature.

---
2. The paper presents its main claims (“valid bounds”, “sharpened partial identification”, etc) early as if fairly general (Sections 1–3). But when you dig into Section 4, you find that the proofs are only in the linear Gaussian “Example 1” setting with heavy structure/assumptions. This can be quite misleading. The authors should either restrict their claims explicitly in the first few sections, or lift their proofs to more general settings (if they can).

---
3. The manuscript repeatedly introduces notation and small definitions close together; readers must hunt back and re-read definitions frequently. I believe this paper can be presented in a much better (readable, at the least) way if a bit of time is spent on it.

---
4. I am not convinced by the idea of using the background knowledge (say symmetries of f) in the way that this paper suggests. Read my question below too.

**Questions:**

If you already have symmetry knowledge about f, why do DA instead of directly imposing constraints?
If the researcher truly knows those symmetries, you can (in many cases) impose them directly in the inference/optimization (e.g., enforce invariance in hypothesis class H, add equality constraints, or augment the objective with invariance penalties) rather than applying DA as a pre-processing step. The manuscript should either 1) clearly justify why DA is preferable to directly injecting invariance into the estimator (computational simplicity? easier to combine with off-the-shelf solvers? better finite-sample behavior?), and provide a brief theoretical or experimental comparison; or 2) explicitly treat DA as one practical way to operationalize symmetry knowledge and discuss tradeoffs (what you lose/gain compared to e.g. constraining H).

---

> ### Author Response · Authors · 2025-11-28
> **Response to Reviewer ba8S**
>
> We thank Reviewer `R. ba8S` for their candid feedback and close reading of the related literature. We took your critique regarding the distinction from prior work and the motivation of DA versus explicit constraints very seriously. This feedback prompted a major restructuring of the paper—specifically the addition of `Sec. 4.1` *(Explicit Constraints)* and a reorganization around `Assumptions 1, 2`—which we believe directly answers your core concerns. We detail these together with other commentary.
>
> # W1. Prior Work, Extension and Novelty
>
> While we found the critical assessment quite valuable, we respectfully disagree with several characterisations and elaborate on these below with an updated draft to highlight the same.
>
> **Attribution and Presentation:**
>
> To clarify, `Prop. 2`, `Lem. 3, 4` are *tools* and **not** *contributions*, and as such restricted solely to the Appendix with no explicit invocation within the main text. However, we acknowledge that lack of explicit attribution may have caused confusion. In the revised manuscript:
>
> - We make attribution to `Akbar et al., 2025` explicit wherever necessary (`Prop. 1`, `Lem. 6`, `Sec. 6.1`).
> - We replaced `Example 1` with a cleaner decomposition: `Assumption 1` (*unbiased-DA*) and `Assumption 2` (linear, Gaussian model).
> - Updated the presentation of the corresponding visualization in `Fig. 2` for distinction.
> - Per `R. 1PwZ`'s remark, we explicitly quantify bounds in `Prop. 1` (*est.-error*) and move it into the main text (`Sec. 3`) to highlight the lack of identifiability, thereby motivating our PI approach.
>
> **Contribution and Novelty:**
>
> **TLDR;** Different problems entirely
>
> We respectfully assert that characterizing this work as a trivial extension of `Akbar et al., 2025` overlooks the fundamental difference in problem setting (*Partial Identification* vs. *Point-Estimation*) and the new objects, tools and machinery at play. While we utilize similar linear-algebraic tools (as one would in *any* linear, Gaussian analysis), the **objects of study**, **metrics**, and **mechanisms of improvement** are distinct.
>
> | Aspect | Akbar et al., 2025 | Our Work |
> |--------|-------------------|----------|
> | **Goal** | point-estimation (in non-identifiable setting) | set recovery (**new**, the standard approach in non-identifiable cases) |
> | **Metrics** | estimation-error | approx.-error, minimax-excess-risk, set volume (all **new**) |
> | **Key Insights** | Symmetry improves point-estimation (`Prop. 1`) | Symmetry improves bound via: (1) *Sharpness* (**new** `Prop. 2`), (2) *Validity* (**new** `Thm. 2`), (3) *Robustness* (**new** `Thm. 1`) |
> | **Mechanisms of Improvement** | *Single:* bias reduction (`Prop. 1`) | *Dual (see `Thm. 1`):* bias reduction (`Prop. 1`) + bound sharpening (**new** `Prop. 2`) |
> | **Tools of Investigation** | change of basis (`Prop. 1`), congruence (`Lem. 7`) | change of basis (`Prop. 1, 2`, `Lem. 6`), congruence (`Lem. 7`), ellipsoidal geometry (**new** `Lem. 2, 3, 4`), Gaussian rotation invariance (**new** `Lem. 5`) |
>
> We believe applying established tools to solve a fundamentally different problem (set recovery) with new metrics and deriving new properties (sharpness/validity/robustness) constitutes a meaningful and novel contribution. To highlight the ones we found particularly interesting:
>
> - **Insights on new mechanisms:** Importantly, our analysis establishes est.-error (`Prop. 1`) and sharper bounds (`Prop. 2`) as *two separate, but related mechanisms* by which symmetry improves PI, as we now explicitly highlight in `Thm. 1` (*robust-bounds*). So **est.-error by itself does *not* give the full picture** for why symmetry may improve PI.
> - **Immediateness of results:** We do not believe the results are immediate. In fact, we find the approx.-error result (`Thm. 2` *(valid bounds)*) in particular to be rather non-intuitive together with `Prop 2` *(sharp bounds)*—**set bounds shrink, yet the boundary is *always* closer to truth?** This was somewhat surprising for us since these are *opposite forces* when $\mathbf{f}\notin \mathcal{H}_{pi}$.
>
> We hope these clarifications and the updated draft suffices for the reviewer to recognise the merit of our work as not merely re-hashing of old ideas, but a meaningful extension of the same.

---

> > ### Author Response · Authors · 2025-11-28
> > **Continued; Response to Reviewer ba8S**
> >
> > # W2. Overly General Early Claims
> >
> > **Fix#1:** Linear-Gaussian setting now explicitly stated in:
> >
> > - Abstract (`line#23`)
> > - Introduction (`line#76`)
> > - Dedicated `Assumption 2` block (`Sec. 3`) instead of potentially misleading “example” block.
> > - All theorem statements reference specific assumptions
> >
> > **Fix#2:** Added explicit *inv.-error constraint* (`Sec. 4.1`) which is applicable more generally, preserving the paper’s relevance beyond linear, Gaussian cases.
> >
> > # W3. Readability
> >
> > Following you and `1PwZ` ’s feedback about symmetry-constraints and concerns about over-emphasis on DA, we realized both the narrative and notation of the paper can be improved significantly.
> >
> > - Abandoned the more confusing notation (e.g., the funny $\operatorname{do}(X\coloneqq GX)$).
> > - Restructured around `Assumptions 1, 2` (instead of `Example 1`)
> > - Moved key `Lemma 2` into main text for better intuition (requested by `R. 1PwZ`)
> > - Re-organized around a symmetry-constraint first narrative (`Weakness 4`)
> >
> > We also remain open to add a notation table in the appendix should the reviewer find it necessary.
> >
> > # W4.+Q1. Operationalising Symmetry Constraints
> >
> > We address this concern by **(a)** introducing an explicit inv.-error constraint in `Sec. 4.1` and **(b)** making the following commentary in comparison to DA pre-processing of `Sec. 4.2`:
> >
> > **Direct constraints (`Sec. 4.1`):**
> >
> > ✓ **General**—sharper by construction (`Remark 1`)
> >
> > ✓ Allows sensitivity analysis via $\epsilon$ parameter (`line#257`)
> >
> > ✓ Works with general PI solvers, including Monte Carlo, gradient-based (`line#243`, `line#260`)
> >
> > ✗ Requires adding constraints to solver (`line#267`)
> >
> > - Implementation cost
> > - Precludes black-box PI software
> >
> > ✗ For complex data, constraint is non-convex ⇒  potentially expensive/unstable (`line#270`)
> >
> > **DA preprocessing (`Sec. 4.2`):**
> >
> > ✓ **Simple**—no solver modification needed (`line#269`)
> >
> > ✓ Computationally cheaper vs. explicit constraint (`line#270`)
> >
> > ✓ **Composable** with standard off-the-shelf PI methods (`line#269`)
> >
> > ✗ Requires well-specified symmetries (though, perhaps possible to account for this by relaxing baseline PI constraints ${\mathbf{\Gamma}}$?).
> >
> > **Their Connection (`line#254`)**: The asymptotic "Large DA" regime (`line#227`) approaches exact constraint results, providing a theoretical bridge between the two approaches. We believe this should directly address your concerns about `Weakness 4` —*”I am not convinced by the idea… in the way that this paper suggests.”*
> >
> > Following this and `R. 1PwZ`'s remarks, we shall be changing the title of the manuscript as below:
> >
> > > ***Symmetry-Constrained Causal Partial Identification***
> >
> > Finally, we thank the reviewer for their feedback, which we believe has drastically improved especially the narrative of the manuscript. Thanks!

---

### Official Review · Reviewer_1PwZ · 2025-10-31

**Soundness:** 3
**Presentation:** 3
**Contribution:** 2
**Rating:** 6
**Confidence:** 4

**Summary:**

This paper studies partial identification bounds under known symmetries of the causal effect, i.e., under invariances of $f$ when $Y = f(X) + \xi$ when $\xi$ may be correlated with $X$, and hence acts as an unobserved confounder. Primarily focusing on the population setting, they define a hypothesis class $H_{pi}$ of possible causal effects functions (each $h \in H_{pi}$ is a function from $x$ to $y$), which are consistent with the observational distribution $P_{X,Y}$ and satisfy the invariance constraints. For each $x$, this hypothesis class induces a set of possible causal effects $H_{pi}(x)$, which in turn can be used to define a worst-case excess risk. Assuming a multivariate Gaussian distribution over $(X, Y, \xi)$, they show that the excess risk strictly decreases under these constraints, as long as the symmetries are not almost surely orthogonal to the expectation of $X$ given $\xi$.

In practice, rather than strictly enforcing the invariances, they are captured using data augmentation. This approach is corroborated through a simulation experiment on a linear model, giving sharper identification bounds, and in real-world experiments on the Optical Device dataset.

**Strengths:**

**Originality and significance:** The use of known symmetries to sharpen partial identification bounds is an interesting and (to the best of my knowledge) novel direction. As the authors discuss, such symmetries are common in many applications, especially in scientific machine learning, where causal inference is quite important, so the combination of the two is a very good match.

**Quality and clarity:** The work is well-executed, the motivation is clear, and the mathematical details are well-written.

**Weaknesses:**

## Major weaknesses

1. **Limitation to multivariate Gaussian setting:** To the best of my understanding, the results are limited to multivariate Gaussian distributions on $(X, Y, \xi)$, and this limitation is not made as transparent as it should be. Based on the text after Assumption 1, I understand that the partial R-squared sensitivity model is not a necessary restriction, but I'm less certain about the Gaussianity assumption (or at least, a linearity assumption). For example, Proposition 1 invokes a Lebesgue measure over $H_{pi}$, which is initially defined as a function space, but in the proofs, $h$ is taken to be a vector (i.e., the coefficients of a linear function). Overall, this lack of transparency gives the feeling that the results are being oversold.
2. **Not enough focus on the quantitative form of the results:** In connection to Weakness 1, I would be much more interested to see Lemma 5 in the main paper and a more quantitative discussion of *how much* the invariances sharpen the partial identification bounds. Theorem 1, Proposition 2, and Theorem 2 don't provide any intuition about how much the invariance sharpens the bounds, which I think would be the most interesting part.

## Minor weakness
3. **Overly focused on data augmentation:** I think a more logical way to present the results would be to focus on how known symmetries/invariances improve the partial identification bound, and afterwards connect these results to data augmentation. The results are really about the restriction of the hypothesis space, and would hold even when strictly enforcing the invariance - the approach of using data augmentation is more of a practical implementation detail.
-

**Questions:**

Please address the Major Weaknesses, especially (1).

---

> ### Author Response · Authors · 2025-11-28
> **Response to Reviewer R. 1PwZ**
>
> We thank the reviewer for their thoughtful comments. While addressing them, we uncovered several additional results that directly strengthen the paper. Your feedback played an important role in this improvement, and we are sincerely grateful.
>
> ---
>
> ## **W1. Linear Gaussian Setting**
>
> **TLDR.** We have addressed concerns about the transparency of the linear–Gaussian assumption by explicitly stating it in
>
> **(1)** the abstract (`line#023`),
>
> **(2)** introduction (`line#076`), and
>
> **(3)** a standalone `Assumption 2` block in `Sec. 3`.
>
> Data augmentation now appears separately under the more general `Assumption 1` in `Sec. 2.4`.
>
> We retain the use of the Lebesgue measure $|\cdot|$ to unify interval width and ellipsoid volume notation.
>
> Finally, motivated by *`Weakness 3`*, we added an **invariance-error–based constraint** in `Sec. 4.1`, which we believe clarifies the relevance of the framework beyond the linear case.
>
> **Extended Comment.** We agree that linearity is a strong and somewhat restrictive assumption. However, it allows us to avoid invoking even stronger structural assumptions—e.g., intuitively we had hoped it would be easy to lift our results to linearity within an RKHS feature space, but we could not get around certain assumptions about feature covariances that are analogous to `Lem. 1` in in the updated manuscript.
>
> It remains plausible that one could bypass function spaces $\mathcal{H}\_{\operatorname{pi}}$ entirely and work directly with distributions $\mathcal{Q}\_{\operatorname{pi}}$ to bypass such limitations, but I have not yet explored this direction sufficiently to make firm claims; a promising direction for future work nonetheless.
>
> ---
>
> ## **W2. Quantitative Results**
>
> **TLDR.** In response to the request for more quantitative statements, we made the following changes:
>
> - The previous `Lem. 5` (now `Lem. 2`) has been moved to `Sec. 4.2` of the main text.
> - We now provide explicit upper and lower bounds on improvement in `Prop. 1` (*est.-error*).
> - We give an explicit quantification for `Prop. 2` (*sharp bounds*), which follows from `Lem. 2` but was important to articulate clearly.
> - For `Thm. 1` (*robust bounds*), we do not provide a standalone quantification; instead, we decompose the result into two interpretable factors:
>     1. an est.-error term (connected to `Prop. 1`), and
>     2. a sharper-bounds term (connected to `Prop. 2`),
>
>         from which **quantitative improvement cascades** directly.
>
> - For `Thm. 2` (*valid bounds*), we do not provide quantitative rates, since **validity**, *not approximation error*, is the central requirement in partial identification.
>
> **Extended Comment.** We agree that explicit quantification strengthens the exposition. Incorporating it has meaningfully improved the narrative:
>
> - `Prop. 1` (*est.-error*) shows directly how **non-identifiability manifests**, directly motivating partial identification and our approach.
> - `Prop. 1` (*est.-error*)and `Prop. 2` (sharp bounds) make clear that **exogenous variation created by DA (**`Lem. 1`**)** is the mechanism driving both estimation improvement and the sharpening of bounds.
> - `Thm. 1` (*robust bounds*) highlights that worst-case error improves through **two complementary mechanisms**—estimation (`Prop. 1`) and sharper bounds (`Prop. 2`).
> - The “large-DA’’ asymptotics (`line#227`) naturally connect with the explicit **invariance-constraint** formulation in `Sec. 4.1`, clarifying how DA approximates the idealized constraint in the limit.
>
> ---
>
> ## **W3. Data-Augmentation Focus**
>
> We incorporated a standalone **invariance-constraint method** in `Sec. 4.1`, and we explicitly note in `line#254`that the **large-DA** regime (`line#227`) corresponds to the idealized limit of this constraint.
>
> We have also reorganized the paper around a **“symmetry-constrained hypothesis class’’** perspective, with DA understood as one concrete mechanism to induce such symmetry.
>
> If accepted, we plan to update the title to:
>
> ***Symmetry-Constrained Causal Partial Identification***
>
> Thank you again for the helpful suggestions—they significantly improved the clarity and direction of the work.

---

### Comment · Reviewer_1PwZ · 2025-11-23
**Authors - please respond to reviews**

To the authors: can you please respond to these reviews? The end of the discussion period is in 10 days and it would be beneficial if the reviewers have time to engage with your responses.

---

> ### Author Response · Authors · 2025-11-28
> **Apologies for the delay!**
>
> We have now posted our responses and welcome further discussion.

---

### Meta-Review · Area_Chair_F54J · 2026-01-05

**Summary:**

The main points of criticism raised by the reviewers are:

- Unclear novelty: Striking similarity to Akbar et al [1], some parts copied without even rewording

- Overly General Early Claims: Results are valid only under highly restrictive assumptions (linearity, multivariate Gaussianity)

- G-invariance is a non-testable assumption.

- Unclear applicability, missing concrete real-world application examples.

- Missing quantitative results

**Reviewer Concerns:**

- Unclear novelty: After the rebuttal the similarities and differences to (Akbar et al.) have become much clearer, but in my opinion, novelty is still rather limited.

- Overly General Early Claims: Results are valid only under highly restrictive assumptions: The crucial dependence on the Gaussian assumption is now stated in a more transparent way in the text, but the original problem could not be addressed.

- G-invariance is a non-testable assumption. The authors agree that this is a indeed a problem. They further explain that essentially all PI methods require such non-testable assumptions. This latter statement is probably true, but doesn't solve the initial problem...

- Unclear applicability, missing concrete real-world application examples. This problem could not be addressed in a convincing way in the rebuttal, too: The only "concrete" example mentioned is "Colored MNIST".

- Missing quantitative results: not addressed in the rebuttal.

**Reviewer Scores:**

From the rebuttal, I don't see many reasons for the reviewers to change their scores significantly.

---

### Decision · Program_Chairs · 2026-01-26

Reject